# Finite-Length Analyses for Source and Channel Coding on Markov Chains[note 1]

**DOI:** 10.3390/e22040460

**Published:** 2020-04-18

**Authors:** Masahito Hayashi, Shun Watanabe

**Affiliations:** 1Shenzhen Institute for Quantum Science and Engineering, Southern University of Science and Technology, Shenzhen 518055, China; 2Graduate School of Mathematics, Nagoya University, Nagoya 464-8602, Japan; 3Center for Quantum Computing, Peng Cheng Laboratory, Shenzhen 518000, China; 4Centre for Quantum Technologies, National University of Singapore, 3 Science Drive 2, Singapore 117542, Singapore; 5Department of Computer and Information Sciences, Tokyo University of Agriculture and Technology, Koganei-shi, Tokyo 184-8588, Japan; shunwata@cc.tuat.ac.jp

**Keywords:** channel coding, Markov chain, finite-length analysis, source coding

## Abstract

We derive finite-length bounds for two problems with Markov chains: source coding with side-information where the source and side-information are a joint Markov chain and channel coding for channels with Markovian conditional additive noise. For this purpose, we point out two important aspects of finite-length analysis that must be argued when finite-length bounds are proposed. The first is the asymptotic tightness, and the other is the efficient computability of the bound. Then, we derive finite-length upper and lower bounds for the coding length in both settings such that their computational complexity is low. We argue the first of the above-mentioned aspects by deriving the large deviation bounds, the moderate deviation bounds, and second-order bounds for these two topics and show that these finite-length bounds achieve the asymptotic optimality in these senses. Several kinds of information measures for transition matrices are introduced for the purpose of this discussion.

## 1. Introduction

In recent years, finite-length analyses for coding problems have been attracting considerable attention [1]. This paper focuses on finite-length analyses for two representative coding problems: One is source coding with side-information for Markov sources, i.e., the Markov–Slepian–Wolf problem on the system $X^{n}$ with full side-information $Y^{n}$ at the decoder, where only the decoder observes the side-information and the source and the side-information are a joint Markov chain. The other is channel coding for channels with Markovian conditional additive noise. Although the main purpose of this paper is finite-length analyses, we also present a unified approach we developed to investigate these topics including asymptotic analyses. Since this discussion is spread across a number of subtopics, we explain them separately in the Introduction.

### 1.1. Two Aspects of Finite-Length Analysis

We explain the motivations of this research by starting with two aspects of finite-length analysis that must be argued when finite-length bounds are proposed. For concreteness, we consider channel coding here even though the problems treated in this paper are not restricted to channel coding. To date, many types of finite-length achievability bounds have been proposed. For example, Verdú and Han derived a finite-length bound by using the information-spectrum approach in order to derive the general formula [2] (see also [3]), which we term as the information-spectrum bound. One of the authors and Nagaoka derived a bound (for the classical-quantum channel) by relating the error probability to binary hypothesis testing [4] (Remark 15) (see also [5]), which we refer to as the hypothesis-testing bound. Polyanskiy et. al. derived the random coding union (RCU) bound and the dependence testing (DT) bound [1] (a bound slightly looser (coefficients are worse) than the DT bound can be derived from the hypothesis-testing bound of [4]). Moreover, Gallager’s bound [6] is known as an efficient bound to derive the exponentially decreasing rate.

Here, we focus on two important aspects of finite-length analysis:**(A1)** Computational complexity for the bound and**(A2)** Asymptotic tightness for the bound.

Both aspects are required for the bound in finite-length analysis as follows. As the first aspect, we consider the computational complexity for the bound. For the BSC (binary symmetric channel), the computational complexity of the RCU bound is $O(n^{2})$, and that of the DT bound is O(n) [7]. However, the computational complexities of these bounds are much larger for general DMCs (discrete memoryless channels) or channels with memory. It is known that the hypothesis testing bound can be described as a linear programming problem (e.g., see [8,9] (in the the case of a quantum channel, the bound is described as a semi-definite programming problem)) and can be efficiently computed under certain symmetry. However, the number of variables in the linear programming problem grows exponentially with the block length, and it is difficult to compute in general. The computation of the information-spectrum bound depends on the evaluation of the tail probability. The hypothesis testing bound gives a tighter bound than the information-spectrum bound, as pointed out by [8], and the computational complexity of the former is much smaller than that of the latter. However, the computation of the tail probability continues to remain challenging unless the channel is a DMC. For DMCs, the computational complexity of Gallager’s bound is O(1) since the Gallager function is an additive quantity for DMCs. However, this is not the case if there is a memory (the Gallager bound for finite-state channels was considered in [10] (Section 5.9), but a closed form expression for the exponent was not derived). Consequently, no efficiently computable bound currently exists for channel coding with Markov additive noise. The situation is the same for source coding with side-information.

Since the actual computation time may depend on the computational resource we can use for numerical experiment, it is not possible to provide a concrete requirement of computational complexity. However, in order to conduct a numerical experiment for a meaningful blocklength, it is reasonable to require the computational complexity to be, at most, a polynomial order of the blocklength *n*.

Next, let us consider the second aspect, i.e., asymptotic tightness. Thus far, three kinds of asymptotic regimes have been studied in information theory [1,11,12,13,14,15,16]:A large deviation regime in which the error probability ε asymptotically behaves as $e^{-nr}$ for some r>0;A moderate deviation regime in which ε asymptotically behaves as $e^{-n^{1-2t}r}$ for some r>0 and t∈(0,1/2); andA second-order regime in which ε is a constant.

We shall claim that a good finite-length bound should be asymptotically optimal for at least one of the above-mentioned three regimes. In fact, the information-spectrum bound, the hypothesis-testing bound, and the DT bound are asymptotically optimal in both the moderate deviation and second-order regimes, whereas the Gallager bound is asymptotically optimal in the large deviation regime and the RCU bound asymptotically optimal in all the regimes (Both the Gallager and RCU bounds are asymptotically optimal in the large deviation regime only up to the critical rate). Recently, for DMCs, Yang and Meng derived an efficiently computable bound for low-density parity check (LDPC) codes [17], which is asymptotically optimal in both the moderate deviation and second-order regimes.

### 1.2. Main Contribution for Finite-Length Analysis

We derive the finite-length achievability bounds on the problems by basically using the exponential-type bounds (for channel coding, it corresponds to the Gallager bound.). In source coding with side-information, the exponential-type upper bounds on error probability ${\bar{P}}_{e}\left(M_{n}\right)$ for a given message size $M_{n}$ are described by using the conditional Rényi entropies as follows (cf. Lemmas 14 and 15):(1)$$\begin{array}{c} {\bar{P}}_{e}\left(M_{n}\right)\leq \underset{-\frac{1}{2}\leq \theta \leq 0}{\inf}M_{n}^{\frac{\theta }{1+\theta }}e^{-\frac{\theta }{1+\theta }H_{1+\theta }^{↑}\left(X^{n}|Y^{n}\right)} \end{array}$$
and:(2)$$\begin{array}{c} {\bar{P}}_{e}\left(M_{n}\right)\leq \underset{-1\leq \theta \leq 0}{\inf}M_{n}^{\theta }e^{-\theta H_{1+\theta }^{↓}\left(X^{n}|Y^{n}\right)}. \end{array}$$

Here, $X^{n}$ is the information to be compressed and $Y^{n}$ is the side-information that can be accessed only by the decoder. $H_{1+\theta }^{↑}\left(X^{n}|Y^{n}\right)$ is the conditional Rényi entropy introduced by Arimoto [18], which we shall refer to as the upper conditional Rényi entropy (cf. (Equation 12)). On the other hand, $H_{1+\theta }^{↓}\left(X^{n}|Y^{n}\right)$ is the conditional Rényi entropy introduced in [19], which we shall refer to as the lower conditional Rényi entropy (cf. (Equation 7)). Although there are several other definitions of conditional Rényi entropies, we only use these two in this paper; see [20,21] for an extensive review on conditional Rényi entropies.

Although the above-mentioned conditional Rényi entropies are additive for i.i.d. random variables, they are not additive for joint Markov chains over $X^{n}$ and $Y^{n}$, for which the derivation of finite-length bounds for Markov chains are challenging. Because it is generally not easy to evaluate the conditional Rényi entropies for Markov chains, we consider two assumptions in relation to transition matrices: the first assumption, which we refer to as non-hidden, is that the *Y*-marginal process is a Markov chain, which enables us to derive the single-letter expression of the conditional entropy rate and the lower conditional Rényi entropy rate; the second assumption, which we refer to as strongly non-hidden, enables us to derive the single-letter expression of the upper conditional Rényi entropy rate; see Assumptions 1 and 2 of Section 2 for more detail (Indeed, as explained later, our result on the data compression can be converted to a result on the channel coding for a specific class of channels. Under this conversion, we obtain certain assumptions for channels. As explained later, these assumptions for channels are more meaningful from a practical point of view.). Under Assumption 1, we introduce the lower conditional Rényi entropy for transition matrices $H_{1+\theta }^{↓,W}\left(X|Y\right)$ (cf. (Equation 47)). Then, we evaluate the lower conditional Rényi entropy for the Markov chain in terms of its transition matrix counterpart. More specifically, we derive an approximation:(3)$$\begin{array}{c} H_{1+\theta }^{↓}\left(X^{n}|Y^{n}\right)=nH_{1+\theta }^{↓,W}\left(X|Y\right)+O\left(1\right), \end{array}$$
where an explicit form of the O(1) term is also derived. Using the evaluation (Equation 2) with this evaluation, we obtain finite-length bounds under Assumption 1. Under a more restrictive assumption, i.e., Assumption 2, we also introduce the upper conditional Rényi entropy for a transition matrix $H_{1+\theta }^{↑,W}\left(X|Y\right)$ (cf. (Equation 55)). Then, we evaluate the upper Rényi entropy for the Markov chain in terms of its transition matrix counterpart. More specifically, we derive an approximation:(4)$$\begin{array}{c} H_{1+\theta }^{↑}\left(X^{n}|Y^{n}\right)=nH_{1+\theta }^{↑,W}\left(X|Y\right)+O\left(1\right), \end{array}$$
where an explicit form of the O(1) term is also derived. Using the evaluation (Equation 1) with this evaluation, we obtain finite-length bounds that are tighter than those obtained under Assumption 1. It should be noted that, without Assumption 1, even the conditional entropy rate is challenging to evaluate. For evaluation of the conditional entropy rate of the *X* process given the *Y* process, the assumption of the *X* process being Markov seems to be not helpful. This is the reason why we consider the *Y* process being Markov instead of the *X* process being Markov in this paper.

We also derive converse bounds by using the change of measure argument for Markov chains developed by the authors in the accompanying paper on information geometry [22,23]. For this purpose, we further introduce two-parameter conditional Rényi entropy and its transition matrix counterpart (cf. (Equation 18) and (Equation 59)). This novel information measure includes the lower conditional Rényi entropy and the upper conditional Rényi entropy as special cases. We clarify the relation among bounds based on these quantities by numerically calculating the upper and lower bounds for the optimal coding rate in source coding with a Markov source in Section 3.7. Owing to the second aspect (A2), this calculation shows that our finite-length bounds are very close to the optimal value. Although this numerical calculation contains a case with a very large size $n=1\times 10^{5}$, its calculation is not as difficult because the calculation complexity behaves as O(1). That is, this calculation shows the advantage of the first aspect (A1).

Here, we would like to remark about the terminologies because there are a few ways to express exponential-type bounds. In statistics or large deviation theory, we usually use the cumulant generating function (CGF) to describe exponents. In information theory, we employ the Gallager function or the Rényi entropies. Although these three terminologies are essentially the same quantity and are related by the change of variables, the CGF and the Gallager function are convenient for some calculations because of their desirable properties such as convexity. On the other hand, the minimum entropy and collision entropy are often used as alternative information measures of Shannon entropy in the community of cryptography. Since the Rényi entropies are a generalization of the minimum entropy and collision entropy, we can regard the Rényi entropies as information measures. The information theoretic meaning of the CGF and the Gallager function are less clear. Thus, the Rényi entropies are intuitively familiar to the readers’ of this journal. The Rényi entropies have an additional advantage in that two types of bounds (e.g., (Equation 152) and (Equation 161)) can be expressed in a unified manner. Therefore, we state our main results in terms of the Rényi entropies, whereas we use the CGF and the Gallager function in the proofs. For the readers’ convenience, the relation between the Rényi entropies and corresponding CGFs are summarized in Appendix A and Appendix B.

### 1.3. Main Contribution for Channel Coding

An intimate relationship is known to exist between channel coding and source coding with side-information (e.g., [24,25,26]). In particular, for an additive channel, the error probability of channel coding by a linear code can be related to the corresponding source coding problem with side-information [24]. Chen et. al. also showed that the error probability of source coding with side-information by a linear encoder can be related to the error probability of a dual channel coding problem and vice versa [27] (see also [28]). Since these dual channels can be regarded as additive channels conditioned on state information, we refer to these channels as conditional additive channels (In [28], we termed these channels general additive channels, but we think “conditional” more suitably describes the situation.). In this paper, we mainly discuss a conditional additive channel, in which the additive noise is operated subject to a distribution conditioned on additional output information. Then, we convert our obtained results of source coding with side-information to the analysis on conditional additive channels. That is, using the aforementioned duality between channel coding and source coding with side-information enables us to evaluate the error probability of channel coding for additive channels. Then, we derive several finite-length analyses on additive channels.

For the same reason as source coding with side-information, we make two assumptions, Assumptions 1 and 2, on the noise process of a conditional additive channel. In this context, Assumption 1 means that the marginal system $Y^{n}$ deciding the behavior of the additive noise $X^{n}$ is a Markov chain. It should be noted that the Gilbert–Elliott channel [29,30] with state information available at the receiver can be regarded as a conditional additive channel such that the noise process is a Markov chain satisfying both Assumptions 1 and 2 (see Example 6). Thus, we believe that Assumptions 1 and 2 are quite reasonable assumptions.

In fact, our analysis is applicable for a broader class of channels known as regular channels [31]. The class of regular channels includes conditional additive channels as a special case, and it is known as a class of channels that are similarly symmetrical. To show it, we propose a method to convert a regular channel into a conditional additive channel such that our treatment covers regular channels. Additionally, we show that the BPSK (binary phase shift keying)-AWGN (additive white Gaussian noise) channel is included in conditional additive channels.

### 1.4. Asymptotic Bounds and Asymptotic Tightness for Finite-Length Bounds

We present asymptotic analyses of the large and moderate deviation regimes by deriving the characterizations (for the large deviation regime, we only derive the characterizations up to the critical rate) with the use of our finite-length achievability and converse bounds, which implies that our finite-length bounds are tight in both of these deviation regimes. We also derive the second-order rate. Although this rate can be derived by the application of the central limit theorem to the information-spectrum bound, the variance involves the limit with respect to the block length because of memory. In this paper, we derive a single-letter form of the variance by using the conditional Rényi entropy for transition matrices (An alternative way to derive a single-letter characterization of the variance for the Markov chain was shown in [32] (Lemma 20). It should also be noted that a single-letter characterization can be derived by using the fundamental matrix [33]. The single-letter characterization of the variance in [12] (Section VII) and [11] (Section III) contains an error, which is corrected in this paper.).

As we will see in Theorems 11–14 and 22–25, our asymptotic results have the same forms as the counterparts of the i.i.d. case (cf. [1,6,11,12,13,14]) when the information measures for distributions in the i.i.d. case are replaced by the information measures for the transition matrices introduced in this paper.

We determine the asymptotic tightness for finite-length bounds by summarizing the relation between the asymptotic results and the finite-length bounds in Table 1. The table also describes the computational complexity of the finite-length bounds. “${\mathrm{Solved}}^{*}$” indicates that those problems are solved up to the critical rates. “Ass. 1” and “Ass. 2” indicate that those problems are solved either under Assumption 1 or Assumption 2. “O(1)” indicates that both the achievability and converse parts of those asymptotic results are derived from our finite-length achievability bounds and converse bounds whose computational complexities are O(1). “Tail” indicates that both the achievability and converse parts of those asymptotic results are derived from the information-spectrum-type achievability bounds and converse bounds of which the computational complexities depend on the computational complexities of the tail probabilities.

In general, the exact computations of tail probabilities are difficult, although they may be feasible for a simple case such as an i.i.d. case. One way to compute tail probabilities approximately is to use the Berry–Esséen theorem [34] (Theorem 16.5.1) or its variant [35]. This direction of research is still ongoing [36,37], and an evaluation of the constant was conducted [37], although its tightness has not been clarified. If we can derive a tight Berry–Esséen-type bound for the Markov chain, this would enable us to derive a finite-length bound that is asymptotically tight in the second-order regime. However, the approximation errors of Berry–Esséen-type bounds converge only in the order of $1/\sqrt{n}$ and cannot be applied when ε is rather small. Even in cases in which the exact computations of tail probabilities are possible, the information-spectrum-type bounds are looser than the exponential type bounds when ε is rather small, and we need to use appropriate bounds depending on the size of ε. In fact, this observation was explicitly clarified in [38] for random number generation with side-information. Consequently, we believe that our exponential-type finite-length bounds are very useful. It should be also noted that, for source coding with side-information and channel coding for regular channels, even the first-order results have not been revealed as far as the authors know, and they are clarified in this paper (General formulae for those problems were known [2,3], but single-letter expressions for Markov sources or channels were not clarified in the literature. For the source coding without side-information, the single-letter expression for entropy rate of Markov source is well known (e.g., see [39]).).

### 1.5. Related Work on Markov Chains

Since related work concerning the finite-length analysis is reviewed in Section 1.1, we only review work related to the asymptotic analysis here. Some studies on Markov chains for the large deviation regime have been reported [40,41,42]. The derivation in [40] used the Markov-type method. A drawback of this method is that it involves a term that stems from the number of types, which does not affect the asymptotic analysis, but does hurt the finite-length analysis. Our achievability is derived by following a similar approach as in [41,42], i.e., the Perron–Frobenius theorem, but our derivation separates the single-shot part and the evaluation of the Rényi entropy, and thus is more transparent. Furthermore, the converse part of [41,42] is based on the Shannon–McMillan–Breiman limiting theorem and does not yield finite-length bounds.

For the second-order regime, Polyanskiy et. al. studied the second-order rate (dispersion) of the Gilbert–Elliott channel [43]. Tomamichel and Tan studied the second-order rate of channel coding with state information such that the state information may be a general source and derived a formula for the Markov chain as a special case [32]. Kontoyiannis studied the second-order variable length source coding for the Markov chain [44]. In [45], Kontoyiannis and Verdú derived the second-order rate of lossless source coding under the overflow probability criterion.

For channel coding of the i.i.d. case, Scarlett et al. derived a saddle-point approximation, which unifies all three regimes [46,47].

### 1.6. Organization of the Paper

In Section 2, we introduce the information measures and their properties that will be used in Section 3 and Section 4. Then, source coding with side-information and channel coding is discussed in Section 3 and Section 4, respectively. As we mentioned above, we state our main result in terms of the Rényi entropies, and we use the CGFs and the Gallager function in the proofs. We explain how to cover the continuous case in Remarks 1 and 5. In Appendix A and Appendix B, the relation between the Rényi entropies and corresponding CGFs are summarized. The relation between the Rényi entropies and the Gallager function are explained as necessary. Proofs of some technical results are also provided in the remaining Appendices.

### 1.7. Notations

For a set X, the set of all distributions on X is denoted by P(X). The set of all sub-normalized non-negative functions on X is denoted by $\bar{P}\left(X\right)$. The cumulative distribution function of the standard Gaussian random variable is denoted by:(5)$$\begin{array}{c} \Phi \left(t\right)=\int_{-\infty }^{t}\frac{1}{\sqrt{2\pi }}\exp\left[-\frac{x^{2}}{2}\right]dx. \end{array}$$

Throughout the paper, the base of the logarithm is the natural base *e*.

## 2. Information Measures

Since this paper discusses the second-order tightness, we need to discuss the central limit theorem for the Markov process. For this purpose, we usually employ advanced mathematical methods from probability theory. For example, the paper [48] (Theorem 4) showed the Markov version of the central limit theorem by using a martingale stopping technique. Lalley [49] employed the regular perturbation theory of operators on the infinite-dimensional space [50] (Chapter 7, #1, Chapter 4, #3, and Chapter 3, #5). The papers [51,52] and [53] (Lemma 1.5 of Chapter 1) employed the spectral measure, while it is hard to calculate the spectral measure in general even in the finite-state case. Further, the papers [36,51,54,55] showed the central limit theorem by using the asymptotic variance, but they did not give any computable expression of the asymptotic variance without the infinite sum. In summary, to derive the central limit theorem with the variance of a computable form, these papers needed to use very advanced mathematics beyond calculus and linear algebra.

To overcome the difficulty of the Markov version of the central limit theorem, we employed the method used in our recent paper [23]. The paper [23] employed the method based on the cumulant generating function for transition matrices, which is defined by the Perron eigenvalue of a specific non-negative-entry matrix. Since a Perron eigenvalue can be explained in the framework of linear algebra, the method can be described with elementary mathematics. To employ this method, we need to define the information measure in a way similar to the cumulant generating function for transition matrices. That is, we define the information measures for transition matrices, e.g., the conditional Rényi entropy for transition matrices, etc, by using Perron eigenvalues.

Fortunately, these information measures for transition matrices are very useful even for large deviation-type evaluation and finite-length bounds. For example, our recent paper [23] derived finite-length bounds for simple hypothesis testing for the Markov chain by using the cumulant generating function for transition matrices. Therefore, using these information measures for transition matrices, this paper derives finite-length bounds for source coding and channel coding with Markov chains and discusses their asymptotic bounds with large deviation, moderate deviation, and the second-order type.

Since they are natural extensions of information measures for single-shot setting, we first review information measures for the single-shot setting in Section 2.1. Next, we introduce information measures for transition matrices in Section 2.2. Then, we show that information measures for Markov chains can be approximated by information measures for transition matrices generating those Markov chains in Section 2.3.

### 2.1. Information Measures for the Single-Shot Setting

In this section, we introduce conditional Rényi entropies for the single-shot setting. For more a detailed review of conditional Rényi entropies, see [21]. For a correlated random variable (X,Y) on X×Y with probability distribution $P_{XY}$ and a marginal distribution $Q_{Y}$ on Y, we introduce the conditional Rényi entropy of order 1+θ relative to $Q_{Y}$ as:(6)$$\begin{array}{c} H_{1+\theta }\left(P_{XY}|Q_{Y}\right):=-\frac{1}{\theta }\log\sum_{x,y}P_{XY}{\left(x,y\right)}^{1+\theta }Q_{Y}{\left(y\right)}^{-\theta }, \end{array}$$
where θ∈(−1,0)∪(0,∞). The conditional Rényi entropy of order zero relative to $Q_{Y}$ is defined by the limit with respect to θ. When *X* has no side-information, it is nothing but the ordinary Rényi entropy, and it is denoted by $H_{1+\theta }\left(X\right)=H_{1+\theta }\left(P_{X}\right)$ throughout the paper.

One of the important special cases of $H_{1+\theta }\left(P_{XY}|Q_{Y}\right)$ is the case with $Q_{Y}=P_{Y}$, where $P_{Y}$ is the marginal of $P_{XY}$. We shall call this special case the lower conditional Rényi entropy of order 1+θ and denote (this notation was first introduced in [56]):(7)$$\begin{array}{ccc} H_{1+\theta }^{↓}\left(X|Y\right) & := & H_{1+\theta }\left(P_{XY}|P_{Y}\right) \end{array}$$(8)$$\begin{array}{cc} = & -\frac{1}{\theta }\log\sum_{x,y}P_{XY}{\left(x,y\right)}^{1+\theta }P_{Y}{\left(y\right)}^{-\theta }. \end{array}$$

When we consider the second-order analysis, the variance of the entropy density plays an important role:(9)$$\begin{array}{c} V\left(X|Y\right):=\mathrm{Var}\left[\log\frac{1}{P_{X|Y}\left(X|Y\right)}\right]. \end{array}$$

We have the following property, which follows from the correspondence between the conditional Rényi entropy and the cumulant generating function (cf. Appendix B).

**Lemma** **1.**
*We have:*
(10)$$\begin{array}{c} \lim_{\theta \to 0}H_{1+\theta }^{↓}\left(X|Y\right)=H\left(X|Y\right) \end{array}$$
*and (as seen in the proof (cf. (Equation 276)), the left-hand side of (Equation 11) corresponds to the second derivative of the cumulant generating function):*
(11)$$\begin{array}{c} \lim_{\theta \to 0}\frac{2\left[H\left(X|Y\right)-H_{1+\theta }^{↓}\left(X|Y\right)\right]}{\theta }=V\left(X|Y\right). \end{array}$$


**Proof.** (Equation 10) follows from the relation in (Equation 275) and the fact that the first-order derivative of the cumulant generating function is the expectation. (Equation 11) follows from (Equation 275), (Equation 10) and (Equation 276). □

The other important special case of $H_{1+\theta }\left(P_{XY}|Q_{Y}\right)$ is the measure maximized over $Q_{Y}$. We shall call this special case the upper conditional Rényi entropy of order 1+θ and denote (Equation (13) for −1<θ<0 follows from the Hölder inequality, and Equation (13) for 0<θ follows from the reverse Hölder inequality [57] (Lemma 8). Similar optimization has appeared in the context of Rényi mutual information in [58] (see also [59]).): (12)$$\begin{array}{ccc} H_{1+\theta }^{↑}\left(X|Y\right) & := & \max_{Q_{Y}\in P\left(Y\right)}H_{1+\theta }\left(P_{XY}|Q_{Y}\right) \end{array}$$(13)$$\begin{array}{cc} = & H_{1+\theta }\left(P_{XY}|P_{Y}^{\left(1+\theta \right)}\right) \end{array}$$
(14)$$\begin{array}{cc} = & -\frac{1+\theta }{\theta }\log\sum_{y}P_{Y}\left(y\right){\left[\sum_{x}P_{X|Y}{\left(x|y\right)}^{1+\theta }\right]}^{\frac{1}{1+\theta }}, \end{array}$$
where:(15)$$\begin{array}{c} P_{Y}^{\left(1+\theta \right)}\left(y\right):=\frac{{\left[\sum_{x}P_{XY}{\left(x,y\right)}^{1+\theta }\right]}^{\frac{1}{1+\theta }}}{\sum_{y^{\prime }}{\left[\sum_{x}P_{XY}{\left(x,y^{\prime }\right)}^{1+\theta }\right]}^{\frac{1}{1+\theta }}}. \end{array}$$

For this measure, we also have the same properties as Lemma 1. This lemma will be proven in Appendix C.

**Lemma** **2.**
*We have:*
(16)$$\begin{array}{c} \lim_{\theta \to 0}H_{1+\theta }^{↑}\left(X|Y\right)=H\left(X|Y\right) \end{array}$$
*and:*
(17)$$\begin{array}{c} \lim_{\theta \to 0}\frac{2\left[H\left(X|Y\right)-H_{1+\theta }^{↑}\left(X|Y\right)\right]}{\theta }=V\left(X|Y\right). \end{array}$$


When we derive converse bounds, we need to consider the case such that the order of the Rényi entropy is different from the order of conditioning distribution defined in (Equation 15). For this purpose, we introduce two-parameter conditional Rényi entropy, which connects the two kinds of conditional Rényi entropies $H_{1+\theta }^{↓}\left(X|Y\right)$ and $H_{1+\theta }^{↑}\left(X|Y\right)$ in the way as Statements 10 and 11 of Lemma 3: (18)$$\begin{array}{c} H_{1+\theta ,1+\theta^{\prime }}\left(X|Y\right) \end{array}$$(19)$$\begin{array}{cc} := & H_{1+\theta }\left(P_{XY}|P_{Y}^{\left(1+\theta^{\prime }\right)}\right) \end{array}$$(20)$$\begin{array}{cc} = & -\frac{1}{\theta }\log\sum_{y}P_{Y}\left(y\right)\left[\sum_{x}P_{X|Y}{\left(x|y\right)}^{1+\theta }\right]{\left[\sum_{x}P_{X|Y}{\left(x|y\right)}^{1+\theta^{\prime }}\right]}^{\frac{-\theta }{1+\theta^{\prime }}}+\frac{\theta^{\prime }}{1+\theta^{\prime }}H_{1+\theta^{\prime }}^{↑}\left(X|Y\right). \end{array}$$

Next, we investigate some properties of the measures defined above, which will be proven in Appendix D.

**Lemma** **3.**
*1*.
*For fixed $Q_{Y}$, $\theta H_{1+\theta }\left(P_{XY}|Q_{Y}\right)$ is a concave function of θ, and it is strict concave iff $\mathrm{Var}\left[\log\frac{Q_{Y}\left(Y\right)}{P_{XY}\left(X,Y\right)}\right]>0$.*
*2*.
*For fixed $Q_{Y}$, $H_{1+\theta }\left(P_{XY}|Q_{Y}\right)$ is a monotonically decreasing (Technically, $H_{1+\theta }\left(P_{XY}|Q_{Y}\right)$ is always non-increasing, and it is monotonically decreasing iff strict concavity holds in Statement 1. Similar remarks are also applied for other information measures throughout the paper.) function of θ.*
*3*.
*The function $\theta H_{1+\theta }^{↓}\left(X|Y\right)$ is a concave function of θ, and it is strict concave iff V(X|Y)>0.*
*4*.
*$H_{1+\theta }^{↓}\left(X|Y\right)$ is a monotonically decreasing function of θ.*
*5*.
*The function $\theta H_{1+\theta }^{↑}\left(X|Y\right)$ is a concave function of θ, and it is strict concave iff V(X|Y)>0.*
*6*.
*$H_{1+\theta }^{↑}\left(X|Y\right)$ is a monotonically decreasing function of θ.*
*7*.
*For every θ∈(−1,0)∪(0,∞), we have $H_{1+\theta }^{↓}\left(X|Y\right)\leq H_{1+\theta }^{↑}\left(X|Y\right)$.*
*8*.
*For fixed $\theta^{\prime }$, the function $\theta H_{1+\theta ,1+\theta^{\prime }}\left(X|Y\right)$ is a concave function of θ, and it is strict concave iff V(X|Y)>0.*
*9*.
*For fixed $\theta^{\prime }$, $H_{1+\theta ,1+\theta^{\prime }}\left(X|Y\right)$ is a monotonically decreasing function of θ.*
*10*.
*We have:*
(21)$$\begin{array}{c} H_{1+\theta ,1}\left(X|Y\right)=H_{1+\theta }^{↓}\left(X|Y\right). \end{array}$$
*11*.
*We have:*
(22)$$\begin{array}{c} H_{1+\theta ,1+\theta }\left(X|Y\right)=H_{1+\theta }^{↑}\left(X|Y\right). \end{array}$$
*12*.
*For every θ∈(−1,0)∪(0,∞), $H_{1+\theta ,1+\theta^{\prime }}\left(X|Y\right)$ is maximized at $\theta^{\prime }=\theta$.*



The following lemma expresses explicit forms of the conditional Rényi entropies of order zero.

**Lemma** **4.**
*We have:*
(23)$$\begin{array}{ccc} \lim_{\theta \to -1}H_{1+\theta }\left(P_{XY}|Q_{Y}\right) & = & H_{0}\left(P_{XY}|Q_{Y}\right) \end{array}$$
(24)$$\begin{array}{cc} := & \log\sum_{y}Q_{Y}\left(y\right)\left|\mathrm{supp}\left(P_{X|Y}\left(\cdot |y\right)\right)\right|, \end{array}$$
(25)$$\begin{array}{ccc} \lim_{\theta \to -1}H_{1+\theta }^{↑}\left(X|Y\right) & = & H_{0}^{↑}\left(X|Y\right) \end{array}$$
(26)$$\begin{array}{cc} := & \log\max_{y\in \mathrm{supp}(P_{Y})}\left|\mathrm{supp}\left(P_{X|Y}\left(\cdot |y\right)\right)\right|, \end{array}$$
(27)$$\begin{array}{ccc} \lim_{\theta \to -1}H_{1+\theta }^{↓}\left(X|Y\right) & = & H_{0}^{↓}\left(X|Y\right) \end{array}$$
(28)$$\begin{array}{cc} := & \log\sum_{y}P_{Y}\left(y\right)\left|\mathrm{supp}\left(P_{X|Y}\left(\cdot |y\right)\right)\right|. \end{array}$$


**Proof.** See Appendix E. □

The definition (Equation 6) guarantees the existence of the derivative of $\frac{d[\theta H_{1+\theta }\left(P_{XY}|Q_{Y}\right)]}{d\theta }$. From Statement 1 of Lemma 3, $d[\theta H_{1+\theta }\left(P_{XY}|Q_{Y}\right)]/d\theta$ is monotonically decreasing. Thus, the inverse function (Throughout the paper, the notations θ(a) and a(R) are reused for several inverse functions. Although the meanings of those notations are obvious from the context, we occasionally put superscript *Q*, ↓ or ↑ to emphasize that those inverse functions are induced from corresponding conditional Rényi entropies. This definition is related to the Legendre transform of the concave function $\theta \mapsto \theta H_{1+\theta }^{↓}\left(X|Y\right)$.) of $\theta \mapsto d[\theta H_{1+\theta }\left(P_{XY}|Q_{Y}\right)]/d\theta$ exists so that the function $\theta \left(a\right)=\theta^{Q}\left(a\right)$ is defined as:(29)$$\begin{array}{c} \frac{d[\theta H_{1+\theta }\left(P_{XY}|Q_{Y}\right)]}{d\theta }{|}_{\theta =\theta (a)}=a \end{array}$$
for $\underline{a}<a\leq \bar{a}$, where $\underline{a}={\underline{a}}^{Q}:=\lim_{\theta \to \infty }d\left[\theta H_{1+\theta }\left(P_{XY}|Q_{Y}\right)\right]/d\theta$ and $\bar{a}={\bar{a}}^{Q}:=\lim_{\theta \to -1}d\left[\theta H_{1+\theta }\left(P_{XY}|Q_{Y}\right)\right]/d\theta$. Let:(30)$$\begin{array}{c} R\left(a\right)=R^{Q}\left(a\right):=\left(1+\theta \left(a\right)\right)a-\theta \left(a\right)H_{1+\theta (a)}\left(P_{XY}|Q_{Y}\right). \end{array}$$

Since:(31)$$\begin{array}{cc} \; & R^{\prime }\left(a\right)=\frac{dR^{\prime }\left(a\right)}{da}\\ = & \frac{d\theta (a)}{da}a+1+\theta \left(a\right)-\frac{d(\theta H_{1+\theta }\left(P_{XY}|Q_{Y}\right))}{d\theta }\frac{d\theta (a)}{da}\\ = & \frac{d\theta (a)}{da}a+1+\theta \left(a\right)-a\frac{d\theta (a)}{da}=1+\theta \left(a\right), \end{array}$$

R(a) is a monotonic increasing function of $\underline{a}<a\leq R\left(\bar{a}\right)$. Thus, we can define the inverse function $a\left(R\right)=a^{Q}\left(R\right)$ of R(a) by:(32)$$\begin{array}{c} \left(1+\theta \left(a\left(R\right)\right)\right)a\left(R\right)-\theta \left(a\left(R\right)\right)H_{1+\theta (a(R))}\left(P_{XY}|Q_{Y}\right)=R \end{array}$$
for $R\left(\underline{a}\right)<R\leq H_{0}\left(P_{XY}|Q_{Y}\right)$.

For $\theta H_{1+\theta }^{↓}\left(X|Y\right)$, by the same reason as above, we can define the inverse functions $\theta \left(a\right)=\theta^{↓}\left(a\right)$ and $a\left(R\right)=a^{↓}\left(R\right)$ by:(33)$$\begin{array}{c} \frac{d[\theta H_{1+\theta }^{↓}\left(X|Y\right)]}{d\theta }{|}_{\theta =\theta (a)}=a \end{array}$$
and:(34)$$\begin{array}{c} \left(1+\theta \left(a\left(R\right)\right)\right)a\left(R\right)-\theta \left(a\left(R\right)\right)H_{1+\theta (a(R))}^{↓}\left(X|Y\right)=R, \end{array}$$
for $R\left(\underline{a}\right)<R\leq H_{0}^{↓}\left(X|Y\right)$. For $\theta H_{1+\theta }^{↑}\left(X|Y\right)$, we also introduce the inverse functions $\theta \left(a\right)=\theta^{↑}\left(a\right)$ and $a\left(R\right)=a^{↑}\left(R\right)$ by:(35)$$\begin{array}{c} \frac{d\theta H_{1+\theta }^{↑}\left(X|Y\right)}{d\theta }{|}_{\theta =\theta (a)}=a \end{array}$$
and:(36)$$\begin{array}{c} \left(1+\theta \left(a\left(R\right)\right)\right)a\left(R\right)-\theta \left(a\left(R\right)\right)H_{1+\theta (a(R))}^{↑}\left(X|Y\right)=R \end{array}$$
for $R\left(\underline{a}\right)<R\leq H_{0}^{↑}\left(X|Y\right)$.

**Remark** **1.**
*Here, we discuss the possibility for extension to the continuous case. Since the entropy in the continuous case diverges, we cannot extend the information quantities to the case when X is continuous. However, it is possible to extend these quantities to the case when Y is continuous, but X is a discrete finite set. In this case, we prepare a general measure μ (like the Lebesgue measure) on Y and probability density function $p_{Y}$ and $q_{Y}$ such that the distributions $P_{Y}$ and $Q_{Y}$ are given as $p_{Y}\left(y\right)\mu \left(dy\right)$ and $q_{Y}\left(y\right)\mu \left(dy\right)$, respectively. Then, it is sufficient to replace ∑, Q(y), and $P_{XY}\left(x,y\right)$ by $\int_{Y}\mu \left(dy\right)$, $P_{X|Y}\left(x|y\right)p_{Y}\left(y\right)$, and $q_{Y}\left(y\right)$, respectively. Hence, in the n-independent and identically distributed case, these information measures are given as n times the original information measures.*

*One might consider the information quantities for transition matrices given in the next subsection for this continuous case. However, this is not so easy because it needs a continuous extension of the Perron eigenvalue.*


### 2.2. Information Measures for the Transition Matrix

Let ${\left\{W\left(x,y|x^{\prime },y^{\prime }\right)\right\}}_{\left(\left(x,y\right),\left(x^{\prime },y^{\prime }\right)\right)\in {\left(X\times Y\right)}^{2}}$ be an ergodic and irreducible transition matrix. The purpose of this section is to introduce transition matrix counterparts of those measures in Section 2.1. For this purpose, we first need to introduce some assumptions on transition matrices:

**Assumption** **1**(Non-hidden). *We say that a transition matrix W is non-hidden (with respect to Y) if the Y-marginal process is a Markov process, i.e., (The reason for the name “non-hidden” is the following. In general, the Y-marginal process is a hidden Markov process. However, when the condition (Equation 37) holds, the Y-marginal process is a Markov process. Hence, we call the condition (Equation 37) non-hidden.):*
(37)$$\begin{array}{c} \sum_{x}W\left(x,y|x^{\prime },y^{\prime }\right)=W\left(y|y^{\prime }\right) \end{array}$$
*for every $x^{\prime }\in X$ and $y,y^{\prime }\in Y$. This condition is equivalent to the existence of the following decomposition of $W(x,y|x^{\prime },y^{\prime })$:*
(38)$$\begin{array}{c} W\left(x,y|x^{\prime },y^{\prime }\right)=W\left(y|y^{\prime }\right)W\left(x|x^{\prime },y^{\prime },y\right). \end{array}$$

**Assumption** **2**(Strongly non-hidden). *We say that a transition matrix W is strongly non-hidden (with respect to Y) if, for every θ∈(−1,∞) and $y,y^{\prime }\in Y$ (The reason for the name “strongly non-hidden” is the following. When we compute the upper conditional Rényi entropy rate of the Markov source, the effect of the Y process may propagate infinitely even if it is non-hidden. When (Equation 39) holds, the effect of the Y process in the computation of the upper conditional Rényi entropy rate is only one step.):*
(39)$$\begin{array}{c} W_{\theta }\left(y|y^{\prime }\right):=\sum_{x}W{\left(x,y|x^{\prime },y^{\prime }\right)}^{1+\theta } \end{array}$$
*is well defined, i.e., the right-hand side of (Equation 39) is independent of $x^{\prime }$.*


Assumption 1 requires (Equation 39) to hold only for θ=0, and thus, Assumption 2 implies Assumption 1. However, Assumption 2 is a strictly stronger condition than Assumption 1. For example, let us consider the case such that the transition matrix is a product form, i.e., $W\left(x,y|x^{\prime },y^{\prime }\right)=W\left(x|x^{\prime }\right)W\left(y|y^{\prime }\right)$. In this case, Assumption 1 is obviously satisfied. However, Assumption 2 is not satisfied in general.

Assumption 2 has another expression as follows.

**Lemma** **5.**
*Assumption 2 holds if and only if, for every $x^{\prime }\neq {\tilde{x}}^{\prime }$, there exists a permutation $\pi_{x^{\prime };{\tilde{x}}^{\prime }}$ on X such that $W\left(x|x^{\prime },y^{\prime },y\right)=W\left(\pi_{x^{\prime };{\tilde{x}}^{\prime }}\left(x\right)|{\tilde{x}}^{\prime },y^{\prime },y\right)$.*


**Proof.** Since the part “if” is trivial, we show the part “only if” as follows. By noting (Equation 38), Assumption 2 can be rephrased as:
(40)$$\begin{array}{c} \sum_{x}W{\left(x|x^{\prime },y^{\prime },y\right)}^{1+\theta } \end{array}$$
does not depend on $x^{\prime }$ for every θ∈(−1,∞). Furthermore, this condition can be rephrased as follows. For $x^{\prime }\neq {\tilde{x}}^{\prime }$, if the largest values of ${\left\{W\left(x|x^{\prime },y^{\prime }\right)\right\}}_{x\in X}$ and ${\left\{W\left(x|{\tilde{x}}^{\prime },y^{\prime }\right)\right\}}_{x\in X}$ are different, say the former is larger, then $\sum_{x}W{\left(x|x^{\prime },y^{\prime }\right)}^{1+\theta }>\sum_{x}W{\left(x|{\tilde{x}}^{\prime },y^{\prime }\right)}^{1+\theta }$ for sufficiently large θ, which contradicts the fact that (Equation 40) does not depend on $x^{\prime }$. Thus, the largest values of ${\left\{W\left(x|x^{\prime },y^{\prime }\right)\right\}}_{x\in X}$ and ${\left\{W\left(x|{\tilde{x}}^{\prime },y^{\prime }\right)\right\}}_{x\in X}$ must coincide. By repeating this argument for the second largest value of ${\left\{W\left(x|x^{\prime },y^{\prime }\right)\right\}}_{x\in X}$ and ${\left\{W\left(x|{\tilde{x}}^{\prime },y^{\prime }\right)\right\}}_{x\in X}$, and so on, we find that Assumption 2 implies that for every $x^{\prime }\neq {\tilde{x}}^{\prime }$, there exists a permutation $\pi_{x^{\prime };{\tilde{x}}^{\prime }}$ on X such that $W\left(x|x^{\prime },y^{\prime },y\right)=W\left(\pi_{x^{\prime };{\tilde{x}}^{\prime }}\left(x\right)|{\tilde{x}}^{\prime },y^{\prime },y\right)$. □

Now, we fix an element $x_{0}\in X$ and transform a sequence of random numbers $\left(X_{1},Y_{1},X_{2},Y_{2},\ldots ,X_{n},Y_{n}\right)$ to the sequence of random numbers $\left(X_{1}^{\prime },Y_{1}^{\prime },X_{2}^{\prime },Y_{2}^{\prime },\ldots ,X_{n}^{\prime },Y_{n}^{\prime }\right):=\left(X_{1},Y_{1},\pi_{x_{0};X_{1}}^{-1}\left(X_{2}\right),Y_{2},\ldots ,\pi_{x_{0};X_{1}}^{-1}\left(X_{n}\right),Y_{n}\right)$. Then, letting $W^{\prime }\left(x|y^{\prime },y\right):=W\left(x|x_{0},y^{\prime },y\right)$, we have $P_{X_{i}^{\prime },Y_{i}^{\prime }|X_{i-1}^{\prime },Y_{i-1}^{\prime }}=W^{\prime }\left(y_{i}^{\prime }|y_{i-1}^{\prime }\right)W\left(x_{i}^{\prime }|y_{i}^{\prime },y_{i-1}^{\prime }\right)$. That is, essentially, the transition matrix of this case can be written by the transition matrix $W\left(y_{i}^{\prime }|y_{i-1}^{\prime }\right)W^{\prime }\left(x_{i}^{\prime }|y_{i}^{\prime },y_{i-1}^{\prime }\right)$. Therefore, the transition matrix can be written by using the positive-entry matrix $W_{x_{i}^{\prime }}\left(y_{i}^{\prime }|y_{i-1}^{\prime }\right):=W\left(y_{i}^{\prime }|y_{i-1}^{\prime }\right)W^{\prime }\left(x_{i}^{\prime }|y_{i}^{\prime },y_{i-1}^{\prime }\right)$.

The following are non-trivial examples satisfying Assumptions 1 and 2.

**Example** **1.**
*Suppose that X=Y is a module (an additive group). Let P and Q be transition matrices on X. Then, the transition matrix given by:*
(41)$$\begin{array}{c} W\left(x,y|x^{\prime },y^{\prime }\right)=Q\left(y|y^{\prime }\right)P\left(x-y|x^{\prime }-y^{\prime }\right) \end{array}$$
*satisfies Assumption 1. Furthermore, if transition matrix $P(z|z^{\prime })$ can be written as:*
(42)$$\begin{array}{c} P\left(z|z^{\prime }\right)=P_{Z}\left(\pi_{z^{\prime }}\left(z\right)\right) \end{array}$$
*for permutation $\pi_{z^{\prime }}$ and a distribution $P_{Z}$ on X, then transition matrix W defined by (Equation 41) satisfies Assumption 2 as well.*


**Example** **2.**
*Suppose that X is a module and W is (strongly) non-hidden with respect to Y. Let Q be a transition matrix on Z=X. Then, the transition matrix given by:*
(43)$$\begin{array}{c} V\left(x,y,z|x^{\prime },y^{\prime },z^{\prime }\right)=W\left(x-z,y|x^{\prime }-z^{\prime },y\right)Q\left(z|z^{\prime }\right) \end{array}$$
*is (strongly) non-hidden with respect to Y×Z.*


The following is also an example satisfying Assumption 2, which describes a noise process of an important class of channels with memory (cf. the Gilbert-Elliot channel in Example 6).

**Example** **3.**
*Let X=Y={0,1}. Then, let:*
(44)$$\begin{array}{c} W\left(y|y^{\prime }\right)=\left\{\begin{array}{cc} 1-q_{y^{\prime }} & ify=y^{\prime }\\ q_{y^{\prime }} & ify\neq y^{\prime } \end{array}\right. \end{array}$$
*for some $0<q_{0},q_{1}<1$, and let:*
(45)$$\begin{array}{c} W\left(x|x^{\prime },y^{\prime },y\right)=\left\{\begin{array}{cc} 1-p_{y} & ifx=0\\ p_{y} & ifx=1 \end{array}\right. \end{array}$$
*for some $0<p_{0},p_{1}<1$. By choosing $\pi_{x^{\prime };{\tilde{x}}^{\prime }}$ to be the identity, this transition matrix satisfies the condition given in Remark 5, which is equivalent to Assumption 2.*


First, we introduce information measures under Assumption 1. In order to define a transition matrix counterpart of (Equation 7), let us introduce the following tilted matrix:(46)$$\begin{array}{c} {\tilde{W}}_{\theta }\left(x,y|x^{\prime },y^{\prime }\right):=W{\left(x,y|x^{\prime },y^{\prime }\right)}^{1+\theta }W{\left(y|y^{\prime }\right)}^{-\theta }. \end{array}$$

Here, we should notice that the tilted matrix ${\tilde{W}}_{\theta }$ is not normalized, i.e., is not a transition matrix. Let $\lambda_{\theta }$ be the Perron–Frobenius eigenvalue of ${\tilde{W}}_{\theta }$ and ${\tilde{P}}_{\theta ,XY}$ be its normalized eigenvector. Then, we define the lower conditional Rényi entropy for *W* by:(47)$$\begin{array}{c} H_{1+\theta }^{↓,W}\left(X|Y\right):=-\frac{1}{\theta }\log\lambda_{\theta }, \end{array}$$
where θ∈(−1,0)∪(0,∞). For θ=0, we define the lower conditional Rényi entropy for *W* by:(48)$$\begin{array}{ccc} H^{W}\left(X|Y\right) & = & H_{1}^{↓,W}\left(X|Y\right) \end{array}$$(49)$$\begin{array}{cc} := & \lim_{\theta \to 0}H_{1+\theta }^{↓,W}\left(X|Y\right), \end{array}$$
and we just call it the conditional entropy for *W*. In fact, the definition of $H^{W}\left(X|Y\right)$ above coincides with:(50)$$\begin{array}{c} -\sum_{x^{\prime },y^{\prime }}P_{0,XY}\left(x^{\prime },y^{\prime }\right)\sum_{x,y}W\left(x,y|x^{\prime },y^{\prime }\right)\log\frac{W(x,y|x^{\prime },y^{\prime })}{W(y|y^{\prime })}, \end{array}$$
where $P_{0,XY}$ is the stationary distribution of *W* (cf. [60] (Equation (Equation 30))). For θ=−1, $H_{0}^{↓,W}\left(X|Y\right)$ is also defined by taking the limit. When *X* has no side-information, the Rényi entropy $H_{1+\theta }^{W}\left(X\right)$ for *W* is defined as a special case of $H_{1+\theta }^{↓,W}\left(X|Y\right)$.

As a counterpart of (Equation 11), we also define (Since the limiting expression in (Equation 51) coincides with the second derivative of the CGF (cf. (Equation 280)) and since the second derivative of the CGF exists (cf. [22] (Appendix D)), the variance in (Equation 51) is well defined. While the definition (Equation 51) contains the limit θ→0, it can be calculated without this type of limit by using the fundamental matrix [61] (Theorem 4.3.1), [23] (Theorem 7.7 and Remark 7.8).):(51)$$\begin{array}{c} V^{W}\left(X|Y\right):=\lim_{\theta \to 0}\frac{2\left[H^{W}\left(X|Y\right)-H_{1+\theta }^{↓,W}\left(X|Y\right)\right]}{\theta }. \end{array}$$

**Remark** **2.**
*When transition matrix W satisfies Assumption 2, $H_{1+\theta }^{↓,W}\left(X|Y\right)$ can be written as:*
(52)$$\begin{array}{c} H_{1+\theta }^{↓,W}\left(X|Y\right)=-\frac{1}{\theta }\log\lambda_{\theta }^{\prime }, \end{array}$$
*where $\lambda_{\theta }^{\prime }$ is the Perron–Frobenius eigenvalue of $W_{\theta }\left(y|y^{\prime }\right)W{\left(y|y^{\prime }\right)}^{-\theta }$. In fact, for the left Perron–Frobenius eigenvector ${\hat{Q}}_{\theta }$ of $W_{\theta }\left(y|y^{\prime }\right)W{\left(y|y^{\prime }\right)}^{-\theta }$, we have:*
(53)$$\begin{array}{c} \sum_{x,y}{\hat{Q}}_{\theta }\left(y\right)W{\left(x,y|x^{\prime },y^{\prime }\right)}^{1+\theta }W{\left(y|y^{\prime }\right)}^{-\theta }=\lambda_{\theta }^{\prime }Q_{\theta }\left(y^{\prime }\right), \end{array}$$
*which implies that $\lambda_{\theta }^{\prime }$ is the Perron–Frobenius eigenvalue of ${\tilde{W}}_{\theta }$. Consequently, we can evaluate $H_{1+\theta }^{↓,W}\left(X|Y\right)$ by calculating the Perron–Frobenius eigenvalue of the |Y|×|Y| matrix instead of the |X||Y|×|X||Y| matrix when W satisfies Assumption 2.*


Next, we introduce information measures under Assumption 2. In order to define a transition matrix counterpart of (Equation 12), let us introduce the following |Y|×|Y| matrix:(54)$$\begin{array}{c} K_{\theta }\left(y|y^{\prime }\right):=W_{\theta }{\left(y|y^{\prime }\right)}^{\frac{1}{1+\theta }}, \end{array}$$
where $W_{\theta }$ is defined by (Equation 39). Let $\kappa_{\theta }$ be the Perron–Frobenius eigenvalue of $K_{\theta }$. Then, we define the upper conditional Rényi entropy for *W* by:(55)$$\begin{array}{c} H_{1+\theta }^{↑,W}\left(X|Y\right):=-\frac{1+\theta }{\theta }\log\kappa_{\theta }, \end{array}$$
where θ∈(−1,0)∪(0,∞). For θ=−1 and θ=0, $H_{1+\theta }^{↑,W}\left(X|Y\right)$ is defined by taking the limit. We have the following properties, which will be proven in Appendix F.

**Lemma** **6.**
*We have:*
(56)$$\begin{array}{c} \lim_{\theta \to 0}H_{1+\theta }^{↑,W}\left(X|Y\right)=H^{W}\left(X|Y\right) \end{array}$$
*and:*
(57)$$\begin{array}{c} \lim_{\theta \to 0}\frac{2\left[H^{W}\left(X|Y\right)-H_{1+\theta }^{↑,W}\left(X|Y\right)\right]}{\theta }=V^{W}\left(X|Y\right). \end{array}$$


Now, let us introduce a transition matrix counterpart of (Equation 18). For this purpose, we introduce the following |Y|×|Y| matrix:(58)$$\begin{array}{c} N_{\theta ,\theta^{\prime }}\left(y|y^{\prime }\right):=W_{\theta }\left(y|y^{\prime }\right)W_{\theta^{\prime }}{\left(y|y^{\prime }\right)}^{\frac{-\theta }{1+\theta^{\prime }}}. \end{array}$$

Let $\nu_{\theta ,\theta^{\prime }}$ be the Perron–Frobenius eigenvalue of $N_{\theta ,\theta^{\prime }}$. Then, we define the two-parameter conditional Rényi entropy by:(59)$$\begin{array}{c} H_{1+\theta ,1+\theta^{\prime }}^{W}\left(X|Y\right):=-\frac{1}{\theta }\log\nu_{\theta ,\theta^{\prime }}+\frac{\theta^{\prime }}{1+\theta^{\prime }}H_{1+\theta^{\prime }}^{↑,W}\left(X|Y\right). \end{array}$$

**Remark** **3.**
*Although we defined $H_{1+\theta }^{↓,W}\left(X|Y\right)$ and $H_{1+\theta }^{↑,W}\left(X|Y\right)$ by (Equation 47) and (Equation 55), respectively, we can alternatively define these measures in the same spirit as the single-shot setting by introducing a transition matrix counterpart of $H_{1+\theta }\left(P_{XY}|Q_{Y}\right)$ as follows. For the marginal $W(y|y^{\prime })$ of $W(x,y|x^{\prime },y^{\prime })$, let $Y_{W}^{2}:=\left\{\left(y,y^{\prime }\right):W\left(y|y^{\prime }\right)>0\right\}$. For another transition matrix V on Y, we define $Y_{V}^{2}$ in a similar manner. For V satisfying $Y_{W}^{2}\subset Y_{V}^{2}$, we define (although we can also define $H_{1+\theta }^{W|V}\left(X|Y\right)$ even if $Y_{W}^{2}\subset Y_{V}^{2}$ is not satisfied (see [22] for the detail), for our purpose of defining $H_{1+\theta }^{↓,W}\left(X|Y\right)$ and $H_{1+\theta }^{↑,W}\left(X|Y\right)$, other cases are irrelevant):*
(60)$$\begin{array}{c} H_{1+\theta }^{W|V}\left(X|Y\right):=-\frac{1}{\theta }\log\lambda_{\theta }^{W|V} \end{array}$$
*for θ∈(−1,0)∪(0,∞), where $\lambda_{\theta }^{W|V}$ is the Perron–Frobenius eigenvalue of:*
(61)$$\begin{array}{c} W{\left(x,y|x^{\prime },y^{\prime }\right)}^{1+\theta }V{\left(y|y^{\prime }\right)}^{-\theta }. \end{array}$$

*By using this measure, we obviously have:*
(62)$$\begin{array}{c} H_{1+\theta }^{↓,W}\left(X|Y\right)=H_{1+\theta }^{W|W}\left(X|Y\right). \end{array}$$

*Furthermore, under Assumption 2, the relation:*
(63)$$\begin{array}{c} H_{1+\theta }^{↑,W}\left(X|Y\right)=\max_{V}H_{1+\theta }^{W|V}\left(X|Y\right) \end{array}$$
*holds (see Appendix G for the proof), where the maximum is taken over all transition matrices satisfying $Y_{W}^{2}\subset Y_{V}^{2}$.*


Next, we investigate some properties of the information measures introduced in this section. The following lemma is proven in Appendix H.

**Lemma** **7.**
*1*.
*The function $\theta H_{1+\theta }^{↓,W}\left(X|Y\right)$ is a concave function of θ, and it is strict concave iff $V^{W}\left(X|Y\right)>0$.*
*2*.
*$H_{1+\theta }^{↓,W}\left(X|Y\right)$ is a monotonically decreasing function of θ.*
*3*.
*The function $\theta H_{1+\theta }^{↑,W}\left(X|Y\right)$ is a concave function of θ, and it is strict concave iff $V^{W}\left(X|Y\right)>0$.*
*4*.
*$H_{1+\theta }^{↑,W}\left(X|Y\right)$ is a monotonically decreasing function of θ.*
*5*.
*For every θ∈(−1,0)∪(0,∞), we have $H_{1+\theta }^{↓,W}\left(X|Y\right)\leq H_{1+\theta }^{↑,W}\left(X|Y\right)$.*
*6*.
*For fixed $\theta^{\prime }$, the function $\theta H_{1+\theta ,1+\theta^{\prime }}^{W}\left(X|Y\right)$ is a concave function of θ, and it is strict concave iff $V^{W}\left(X|Y\right)>0$.*
*7*.
*For fixed $\theta^{\prime }$, $H_{1+\theta ,1+\theta^{\prime }}^{W}\left(X|Y\right)$ is a monotonically decreasing function of θ.*
*8*.
*We have:*
(64)$$\begin{array}{c} H_{1+\theta ,1}^{W}\left(X|Y\right)=H_{1+\theta }^{↓,W}\left(X|Y\right). \end{array}$$
*9*.
*We have:*
(65)$$\begin{array}{c} H_{1+\theta ,1+\theta }^{W}\left(X|Y\right)=H_{1+\theta }^{↑,W}\left(X|Y\right). \end{array}$$
*10*.
*For every θ∈(−1,0)∪(0,∞), $H_{1+\theta ,1+\theta^{\prime }}^{W}\left(X|Y\right)$ is maximized at $\theta^{\prime }=\theta$, i.e.,*
(66)$$\begin{array}{c} \frac{d[H_{1+\theta ,1+\theta^{\prime }}^{W}\left(X|Y\right)]}{d\theta^{\prime }}{|}_{\theta^{\prime }=\theta }=0. \end{array}$$



From Statement 1 of Lemma 7, $d[\theta H_{1+\theta }^{↓,W}\left(X|Y\right)]/d\theta$ is monotonically decreasing. Thus, we can define the inverse function $\theta^{W}\left(a\right)=\theta^{↓,W}\left(a\right)$ of $d[\theta H_{1+\theta }^{↓,W}\left(X|Y\right)]/d\theta$ by:(67)$$\begin{array}{c} \frac{d[\theta H_{1+\theta }^{↓,W}\left(X|Y\right)]}{d\theta }{|}_{\theta =\theta^{W}\left(a\right)}=a \end{array}$$
for $\underline{a}<a\leq \bar{a}$, where $\underline{a}:=\lim_{\theta \to \infty }d\left[\theta H_{1+\theta }^{↓,W}\left(X|Y\right)\right]/d\theta$ and $\bar{a}:=\lim_{\theta \to -1}d\left[\theta H_{1+\theta }^{↓,W}\left(X|Y\right)\right]/d\theta$. Let:(68)$$\begin{array}{c} R^{W}\left(a\right):=\left(1+\theta \left(a\right)\right)a-\theta \left(a\right)H_{1+\theta (a)}^{↓,W}\left(X|Y\right). \end{array}$$

Since
(69)$$\begin{array}{c} {R^{W}}^{\prime }\left(a\right)=\left(1+\theta \left(a\right)\right), \end{array}$$

$R^{W}\left(a\right)$ is a monotonic increasing function of $\underline{a}<a<R^{W}\left(\bar{a}\right)$. Thus, we can define the inverse function $a^{W}\left(R\right)=a^{↓,W}\left(R\right)$ of $R^{W}\left(a\right)$ by:(70)$$\begin{array}{c} \left(1+\theta \left(a^{W}\left(R\right)\right)\right)a^{W}\left(R\right)-\theta^{W}\left(a^{W}\left(R\right)\right)H_{1+\theta^{W}\left(a^{W}\left(R\right)\right)}^{↓,W}\left(X|Y\right)=R \end{array}$$
for $R^{W}\left(\underline{a}\right)<R<H_{0}^{↓,W}\left(X|Y\right)$, where $H_{0}^{↓,W}\left(X|Y\right):=\lim_{\theta \to -1}H_{1+\theta }^{↓,W}\left(X|Y\right)$.

For $\theta H_{1+\theta }^{↑,W}\left(X|Y\right)$, by the same reason, we can define the inverse function $\theta^{W}\left(a\right)=\theta^{↑,W}\left(a\right)$ by:(71)$$\begin{array}{c} \frac{d[\theta H_{1+\theta ,1+\theta^{W}\left(a\right)}^{W}\left(X|Y\right)]}{d\theta }{|}_{\theta =\theta^{W}\left(a\right)}=\frac{d[\theta H_{1+\theta }^{↑,W}\left(X|Y\right)]}{d\theta }{|}_{\theta =\theta^{W}\left(a\right)}=a, \end{array}$$
and the inverse function $a^{W}\left(R\right)=a^{↑,W}\left(R\right)$ of:(72)$$\begin{array}{c} R^{W}\left(a\right):=\left(1+\theta^{W}\left(a\right)\right)a-\theta^{W}\left(a\right)H_{1+\theta^{W}\left(a\right)}^{↑,W}\left(X|Y\right) \end{array}$$
by:(73)$$\begin{array}{c} \left(1+\theta^{W}\left(a^{W}\left(R\right)\right)\right)a^{W}\left(R\right)-\theta^{W}\left(a^{W}\left(R\right)\right)H_{1+\theta^{W}\left(a^{W}\left(R\right)\right)}^{↑,W}\left(X|Y\right)=R, \end{array}$$
for $R\left(\underline{a}\right)<R<H_{0}^{↑,W}\left(X|Y\right)$, where $H_{0}^{↑,W}\left(X|Y\right):=\lim_{\theta \to -1}H_{1+\theta }^{↑,W}\left(X|Y\right)$. Here, the first equality in (Equation 71) follows from (Equation 66).

Since $\theta \mapsto \theta H_{1+\theta }^{↓,W}\left(X|Y\right)$ is concave, the supremum of $\left[-\theta R+\theta H_{1+\theta }^{↓,W}\left(X|Y\right)\right]$ is attained at the stationary point. Furthermore, note that $-1\leq \theta^{↓,W}\left(R\right)\leq 0$ for $H^{W}\left(X|Y\right)\leq R\leq H_{0}^{↓,W}\left(X|Y\right)$. Thus, we have the following property.

**Lemma** **8.**
*The function $\theta^{W}\left(R\right)$ defined in (Equation 67) satisfies:*
(74)$$\begin{array}{c} \underset{-1\leq \theta \leq 0}{\sup}\left[-\theta R+\theta H_{1+\theta }^{↓,W}\left(X|Y\right)\right]=-\theta^{W}\left(R\right)R+\theta^{W}\left(R\right)H_{1+\theta^{W}\left(R\right)}^{↓,W}\left(X|Y\right) \end{array}$$
*for $H^{W}\left(X|Y\right)\leq R\leq H_{0}^{↓,W}\left(X|Y\right)$.*


Furthermore, we have the following characterization for another type of maximization.

**Lemma** **9.**
*The function $\theta^{W}\left(a^{W}\left(R\right)\right)$ defined by (Equation 70) satisfies:*
(75)$$\begin{array}{c} \underset{-1\leq \theta \leq 0}{\sup}\frac{-\theta R+\theta H_{1+\theta }^{↓,W}\left(X|Y\right)}{1+\theta }=-\theta^{W}\left(a^{W}\left(R\right)\right)a^{W}\left(R\right)+\theta^{W}\left(a^{W}\left(R\right)\right)H_{1+\theta^{W}\left(a^{W}\left(R\right)\right)}^{↓,W}\left(X|Y\right) \end{array}$$
*for $H^{W}\left(X|Y\right)\leq R\leq H_{0}^{↓,W}\left(X|Y\right)$, and the function θ(a(R)) defined in (Equation 73) satisfies:*
(76)$$\begin{array}{c} \underset{-1\leq \theta \leq 0}{\sup}\frac{-\theta R+\theta H_{1+\theta }^{↑,W}\left(X|Y\right)}{1+\theta }=-\theta^{W}\left(a^{W}\left(R\right)\right)a^{W}\left(R\right)+\theta^{W}\left(a^{W}\left(R\right)\right)H_{1+\theta^{W}\left(a^{W}\left(R\right)\right)}^{↑,W}\left(X|Y\right) \end{array}$$
*for $H^{W}\left(X|Y\right)\leq R\leq H_{0}^{↑,W}\left(X|Y\right)$.*


**Proof.** See Appendix I. □

**Remark** **4.**
*The combination of (Equation 49), (Equation 51), and Lemma 6 guarantees that both the conditional Rényi entropies expand as:*
(77)$$\begin{array}{ccc} H_{1+\theta }^{↓,W}\left(X|Y\right) & = & H^{W}\left(X|Y\right)-\frac{1}{2}V^{W}\left(X|Y\right)\theta +o\left(\theta \right), \end{array}$$
(78)$$\begin{array}{ccc} H_{1+\theta }^{↑,W}\left(X|Y\right) & = & H^{W}\left(X|Y\right)-\frac{1}{2}V^{W}\left(X|Y\right)\theta +o\left(\theta \right) \end{array}$$

*around θ=0. Thus, the difference of these measures significantly appears only when |θ| is rather large. For the transition matrix of Example 3 with $q_{0}=q_{1}=0.1$, $p_{0}=0.1$, and $p_{1}=0.4$, we plotted the values of the information measures in Figure 1. Although the values at θ=−1 coincide in Figure 1, note that the values at θ=−1 may differ in general.*

*In Example 1, we mentioned that the transition matrix W in (Equation 41) satisfies Assumption 2 when transition matrix P is given by (Equation 42). By computing the conditional Rényi entropies for this special case, we have:*
(79)$$\begin{array}{ccc} H_{1+\theta }^{↑,W}\left(X|Y\right) & = & H_{1+\theta }^{↓,W}\left(X|Y\right) \end{array}$$
(80)$$\begin{array}{cc} = & H_{1+\theta }\left(P_{Z}\right), \end{array}$$
*i.e., the two kinds of conditional Rényi entropies coincide.*


Now, let us consider the asymptotic behavior of $H_{1+\theta }^{↓,W}\left(X|Y\right)$ around θ=0. When θ(a) is close to zero, we have:(81)$$\begin{array}{c} \theta^{W}\left(a\right)H_{1+\theta^{W}\left(a\right)}^{↓,W}\left(X|Y\right)=\theta^{W}\left(a\right)H^{W}\left(X|Y\right)-\frac{1}{2}V^{W}\left(X|Y\right)\theta^{W}{\left(a\right)}^{2}+o\left(\theta^{W}{\left(a\right)}^{2}\right). \end{array}$$

Taking the derivative, (Equation 67) implies that:(82)$$\begin{array}{c} a=H^{W}\left(X|Y\right)-V^{W}\left(X|Y\right)\theta^{W}\left(a\right)+o\left(\theta^{W}\left(a\right)\right). \end{array}$$

Hence, when *R* is close to $H^{W}\left(X|Y\right)$, we have:(83)$$\begin{array}{cc} R\; & =(1+\theta^{W}\left(a^{W}\left(R\right)\right)a^{W}\left(R\right)-\theta^{W}\left(a^{W}\left(R\right)\right)H_{1+\theta^{W}\left(a^{W}\left(R\right)\right)}^{↓,W}\left(X|Y\right) \end{array}$$(84)$$\begin{array}{cc} \; & =H^{W}\left(X|Y\right)-\left(1+\frac{\theta^{W}\left(a^{W}\left(R\right)\right)}{2}\right)\theta^{W}\left(a^{W}\left(R\right)\right)V^{W}\left(X|Y\right)+o(\theta^{W}\left(a^{W}\left(R\right)\right), \end{array}$$
i.e.,
(85)$$\begin{array}{c} \theta^{W}\left(a^{W}\left(R\right)\right)=\frac{-R+H^{W}\left(X|Y\right)}{V^{W}\left(X|Y\right)}+o\left(\frac{R-H^{W}\left(X|Y\right)}{V^{W}\left(X|Y\right)}\right). \end{array}$$

Furthermore, (Equation 81) and (Equation 82) imply:(86)$$\begin{array}{cc} \; & -\theta^{W}\left(a^{W}\left(R\right)\right)a^{W}\left(R\right)+\theta^{W}\left(a^{W}\left(R\right)\right)H_{1+\theta^{W}\left(a^{W}\left(R\right)\right)}^{↓,W}\left(X|Y\right) \end{array}$$(87)$$\begin{array}{cc} \; & =V^{W}\left(X|Y\right)\frac{\theta^{W}{\left(a^{W}\left(R\right)\right)}^{2}}{2}+o\left(\theta^{W}{\left(a^{W}\left(R\right)\right)}^{2}\right) \end{array}$$(88)$$\begin{array}{cc} \; & =\frac{V^{W}\left(X|Y\right)}{2}{\left(\frac{R-H^{W}\left(X|Y\right)}{V^{W}\left(X|Y\right)}\right)}^{2}+o\left({\left(\frac{R-H^{W}\left(X|Y\right)}{V^{W}\left(X|Y\right)}\right)}^{2}\right). \end{array}$$

### 2.3. Information Measures for the Markov Chain

Let (X,Y) be the Markov chain induced by transition matrix *W* and some initial distribution $P_{X_{1}Y_{1}}$. Now, we show how information measures introduced in Section 2.2 are related to the conditional Rényi entropy rates. First, we introduce the following lemma, which gives finite upper and lower bounds on the lower conditional Rényi entropy.

**Lemma** **10.**
*Suppose that transition matrix W satisfies Assumption 1. Let $v_{\theta }$ be the eigenvector of $W_{\theta }^{T}$ with respect to the Perron–Frobenius eigenvalue $\lambda_{\theta }$ such that $\min_{x,y}v_{\theta }\left(x,y\right)=1$ (since the eigenvector corresponding to the Perron–Frobenius eigenvalue for an irreducible non-negative matrix has always strictly positive entries [62] (Theorem 8.4.4, p. 508), we can choose the eigenvector $v_{\theta }$ satisfying this condition). Let $w_{\theta }\left(x,y\right):=P_{X_{1}Y_{1}}{\left(x,y\right)}^{1+\theta }P_{Y_{1}}{\left(y\right)}^{-\theta }$. Then, for every n≥1, we have:*
(89)$$\begin{array}{c} \left(n-1\right)\theta H_{1+\theta }^{↓,W}\left(X|Y\right)+\underline{\delta }\left(\theta \right)\leq \theta H_{1+\theta }^{↓}\left(X^{n}|Y^{n}\right)\leq \left(n-1\right)\theta H_{1+\theta }^{↓,W}\left(X|Y\right)+\bar{\delta }\left(\theta \right), \end{array}$$
*where:*
(90)$$\begin{array}{ccc} \bar{\delta }\left(\theta \right) & := & -\log\langle v_{\theta }|w_{\theta }\rangle +\log\max_{x,y}v_{\theta }\left(x,y\right), \end{array}$$
(91)$$\begin{array}{ccc} \underline{\delta }\left(\theta \right) & := & -\log\langle v_{\theta }|w_{\theta }\rangle , \end{array}$$
*and $\langle v_{\theta }|w_{\theta }\rangle$ is defined as $\sum_{x,y}v_{\theta }\left(x,y\right)w_{\theta }\left(x,y\right)$.*


**Proof.** This follows from (Equation 279) and Lemma A2. □

From Lemma 10, we have the following.

**Theorem** **1.**
*Suppose that transition matrix W satisfies Assumption 1. For any initial distribution, we have (When there is no side-information, (Equation 93) reduces to the well-known expression of the entropy rate of the Markov process [39]. Without Assumption 1, it is not clear if (Equation 93) holds or not.):*
(92)$$\begin{array}{ccc} \lim_{n\to \infty }\frac{1}{n}H_{1+\theta }^{↓}\left(X^{n}|Y^{n}\right) & = & H_{1+\theta }^{↓,W}\left(X|Y\right), \end{array}$$
(93)$$\begin{array}{ccc} \lim_{n\to \infty }\frac{1}{n}H\left(X^{n}|Y^{n}\right) & = & H^{W}\left(X|Y\right). \end{array}$$


We also have the following asymptotic evaluation of the variance, which follows from Lemma A3 in Appendix A.

**Theorem** **2.**
*Suppose that transition matrix W satisfies Assumption 1. For any initial distribution, we have:*
(94)$$\begin{array}{c} \lim_{n\to \infty }\frac{1}{n}V\left(X^{n}|Y^{n}\right)=V^{W}\left(X|Y\right). \end{array}$$


Theorem 2 is practically important since the limit of the variance can be described by a single-letter characterized quantity. A method to calculate $V^{W}\left(X|Y\right)$ can be found in [23].

Next, we show the lemma that gives the finite upper and lower bounds on the upper conditional Rényi entropy in terms of the upper conditional Rényi entropy for the transition matrix.

**Lemma** **11.**
*Suppose that transition matrix W satisfies Assumption 2. Let $v_{\theta }$ be the eigenvector of $K_{\theta }^{T}$ with respect to the Perron–Frobenius eigenvalue $\kappa_{\theta }$ such that $\min_{y}v_{\theta }\left(y\right)=1$. Let $w_{\theta }$ be the |Y|-dimensional vector defined by:*
(95)$$\begin{array}{c} w_{\theta }\left(y\right):={\left[\sum_{x}P_{X_{1}Y_{1}}{\left(x,y\right)}^{1+\theta }\right]}^{\frac{1}{1+\theta }}. \end{array}$$

*Then, we have:*
(96)$$\begin{array}{c} \left(n-1\right)\frac{\theta }{1+\theta }H_{1+\theta }^{↑,W}\left(X|Y\right)+\underline{\xi }\left(\theta \right)\leq \frac{\theta }{1+\theta }H_{1+\theta }^{↑}\left(X^{n}|Y^{n}\right)\leq \left(n-1\right)\frac{\theta }{1+\theta }H_{1+\theta }^{↑,W}\left(X|Y\right)+\bar{\xi }\left(\theta \right), \end{array}$$
*where:*
(97)$$\begin{array}{ccc} \bar{\xi }\left(\theta \right) & := & -\log\langle v_{\theta }|w_{\theta }\rangle +\log\max_{y}v_{\theta }\left(y\right), \end{array}$$
(98)$$\begin{array}{ccc} \underline{\xi }\left(\theta \right) & := & -\log\langle v_{\theta }|w_{\theta }\rangle . \end{array}$$


**Proof.** See Appendix J. □

From Lemma 11, we have the following.

**Theorem** **3.**
*Suppose that transition matrix W satisfies Assumption 2. For any initial distribution, we have:*
(99)$$\begin{array}{ccc} \lim_{n\to \infty }\frac{1}{n}H_{1+\theta }^{↑}\left(X^{n}|Y^{n}\right) & = & H_{1+\theta }^{↑,W}\left(X|Y\right). \end{array}$$


Finally, we show the lemma that gives the finite upper and lower bounds on the two-parameter conditional Rényi entropy in terms of the two-parameter conditional Rényi entropy for the transition matrix.

**Lemma** **12.**
*Suppose that transition matrix W satisfies Assumption 2. Let $v_{\theta ,\theta^{\prime }}$ be the eigenvector of $N_{\theta ,\theta^{\prime }}^{T}$ with respect to the Perron–Frobenius eigenvalue $\nu_{\theta ,\theta^{\prime }}$ such that $\min_{y}v_{\theta ,\theta^{\prime }}\left(y\right)=1$. Let $w_{\theta ,\theta^{\prime }}$ be the |Y|-dimensional vector defined by:*
(100)$$\begin{array}{c} w_{\theta ,\theta^{\prime }}\left(y\right):=\left[\sum_{x}P_{X_{1}Y_{1}}{\left(x,y\right)}^{1+\theta }\right]{\left[\sum_{x}P_{X_{1}Y_{1}}{\left(x,y\right)}^{1+\theta^{\prime }}\right]}^{\frac{-\theta }{1+\theta^{\prime }}}. \end{array}$$

*Then, we have:*
(101)$$\begin{array}{c} \left(n-1\right)\theta H_{1+\theta ,1+\theta^{\prime }}^{W}\left(X|Y\right)+\underline{\zeta }\left(\theta ,\theta^{\prime }\right)\leq \theta H_{1+\theta ,1+\theta^{\prime }}\left(X^{n}|Y^{n}\right)\leq \left(n-1\right)\theta H_{1+\theta ,1+\theta^{\prime }}^{W}\left(X|Y\right)+\bar{\zeta }\left(\theta ,\theta^{\prime }\right), \end{array}$$

*where:*
(102)$$\begin{array}{ccc} \bar{\zeta }\left(\theta ,\theta^{\prime }\right) & := & -\log\langle v_{\theta ,\theta^{\prime }}|w_{\theta ,\theta^{\prime }}\rangle +\log\max_{y}v_{\theta ,\theta^{\prime }}\left(y\right)+\theta \bar{\xi }\left(\theta^{\prime }\right), \end{array}$$
(103)$$\begin{array}{ccc} \underline{\zeta }\left(\theta ,\theta^{\prime }\right) & := & -\log\langle v_{\theta ,\theta^{\prime }}|w_{\theta ,\theta^{\prime }}\rangle +\theta \underline{\xi }\left(\theta^{\prime }\right) \end{array}$$
*for θ>0 and:*
(104)$$\begin{array}{ccc} \bar{\zeta }\left(\theta ,\theta^{\prime }\right) & := & -\log\langle v_{\theta ,\theta^{\prime }}|w_{\theta ,\theta^{\prime }}\rangle +\log\max_{y}v_{\theta ,\theta^{\prime }}\left(y\right)+\theta \underline{\xi }\left(\theta^{\prime }\right), \end{array}$$
(105)$$\begin{array}{ccc} \underline{\zeta }\left(\theta ,\theta^{\prime }\right) & := & -\log\langle v_{\theta ,\theta^{\prime }}|w_{\theta ,\theta^{\prime }}\rangle +\theta \bar{\xi }\left(\theta^{\prime }\right) \end{array}$$
*for θ<0*


**Proof.** By multiplying θ in the definition of $H_{1+\theta ,1+\theta^{\prime }}\left(X^{n}|Y^{n}\right)$, we have:
(106)$$\begin{array}{c} \theta H_{1+\theta ,1+\theta^{\prime }}\left(X^{n}|Y^{n}\right) \end{array}$$
(107)$$\begin{array}{cc} = & -\log\sum_{y^{n}}\left[\sum_{x^{n}}P_{X^{n}Y^{n}}{\left(x^{n},y^{n}\right)}^{1+\theta }\right]{\left[\sum_{x^{n}}P_{X^{n}Y^{n}}{\left(x^{n},y^{n}\right)}^{1+\theta^{\prime }}\right]}^{\frac{-\theta }{1+\theta^{\prime }}}+\frac{\theta \theta^{\prime }}{1+\theta^{\prime }}H_{1+\theta^{\prime }}^{↑}\left(X^{n}|Y^{n}\right). \end{array}$$The second term is evaluated by Lemma 11. The first term can be evaluated almost in the same manner as Lemma 11. □

From Lemma 12, we have the following.

**Theorem** **4.**
*Suppose that transition matrix W satisfies Assumption 2. For any initial distribution, we have:*
(108)$$\begin{array}{c} \lim_{n\to \infty }\frac{1}{n}H_{1+\theta ,1+\theta^{\prime }}\left(X^{n}|Y^{n}\right)=H_{1+\theta ,1+\theta^{\prime }}^{W}\left(X|Y\right). \end{array}$$


## 3. Source Coding with Full Side-Information

In this section, we investigate source coding with side-information. We start this section by showing the problem setting in Section 3.1. Then, we review and introduce some single-shot bounds in Section 3.2. We derive finite-length bounds for the Markov chain in Section 3.3. Then, in Section 3.5 and Section 3.6, we show the asymptotic characterization for the large deviation regime and the moderate deviation regime by using those finite-length bounds. We also derive the second-order rate in Section 3.4.

### 3.1. Problem Formulation

A code Ψ=(e,d) consists of one encoder e:X→{1,…,M} and one decoder d:{1,…,M}×Y→X. The decoding error probability is defined by:(109)$$\begin{array}{ccc} P_{s}\left[\Psi \right] & = & P_{s}\left[\Psi |P_{XY}\right] \end{array}$$(110)$$\begin{array}{cc} := & \mathrm{Pr}\{X\neq d(e(X),Y)\}. \end{array}$$

For notational convenience, we introduce the infimum of error probabilities under the condition that the message size is *M*:(111)$$\begin{array}{ccc} P_{s}\left(M\right) & = & P_{s}\left(M|P_{XY}\right) \end{array}$$(112)$$\begin{array}{cc} := & \underset{\Psi }{\inf}P_{s}\left[\Psi \right]. \end{array}$$

For theoretical simplicity, we focus on a randomized choice of our encoder. For this purpose, we employ a randomized hash function *F* from X to {1,…,M}. A randomized hash function *F* is called a two-universal hash when $\mathrm{Pr}\left\{F\left(x\right)=F\left(x^{\prime }\right)\right\}\leq \frac{1}{M}$ for any distinctive *x* and $x^{\prime }$ [63]; the so-called bin coding [39] is an example of the two-universal hash function. In the following, we denote the set of two-universal hash functions by F. Given an encoder *f* as a function from X to {1,…,M}, we define the decoder $d_{f}$ as the optimal decoder by $\underset{d}{\mathrm{argmin}}P_{s}\left[\left(f,d\right)\right]$. Then, we denote the code $\left(f,d_{f}\right)$ by Ψ(f). Then, we bound the error probability $P_{s}\left[\Psi \left(F\right)\right]$ averaged over the random function *F* by only using the property of two-universality. In order to consider the worst case of such schemes, we introduce the following quantity: (113)$$\begin{array}{ccc} {\bar{P}}_{s}\left(M\right) & = & {\bar{P}}_{s}\left(M|P_{XY}\right) \end{array}$$(114)$$\begin{array}{cc} := & \underset{F\in F}{\sup}E_{F}\left[P_{s}\left[\Psi \left(F\right)\right]\right],. \end{array}$$

When we consider *n*-fold extension, the source code and related quantities are denoted with the superscript (n). For example, the quantities in (Equation 112) and (Equation 114) are written as $P_{s}^{\left(n\right)}\left(M\right)$ and ${\bar{P}}_{s}^{\left(n\right)}\left(M\right)$, respectively. Instead of evaluating them, we are often interested in evaluating:(115)$$\begin{array}{ccc} M(n,\varepsilon ) & := & \inf\{M_{n}:P_{s}^{\left(n\right)}\left(M_{n}\right)\leq \varepsilon \}, \end{array}$$(116)$$\begin{array}{ccc} \bar{M}\left(n,\varepsilon \right) & := & \inf\{M_{n}:{\bar{P}}_{s}^{\left(n\right)}\left(M_{n}\right)\leq \varepsilon \} \end{array}$$
for given 0≤ε<1.

### 3.2. Single-Shot Bounds

In this section, we review existing single-shot bounds and also show novel converse bounds. For the information measures used below, see Section 2.

By using the standard argument on information-spectrum approach, we have the following achievability bound.

**Lemma** **13**(Lemma 7.2.1 of [3]). *The following bound holds:*
(117)$$\begin{array}{c} {\bar{P}}_{s}\left(M\right)\leq \underset{\gamma \geq 0}{\inf}\left[P_{XY}\left\{\log\frac{1}{P_{X|Y}\left(x|y\right)}>\gamma \right\}+\frac{e^{\gamma }}{M}\right]. \end{array}$$

Although Lemma 13 is useful for the second-order regime, it is known to be not tight in the large deviation regime. By using the large deviation technique of Gallager, we have the following exponential-type achievability bound.

**Lemma** **14**([64]). *The following bound holds: (note that the Gallager function and the upper conditional Rényi entropy are related by (Equation 295)):*
(118)$$\begin{array}{c} {\bar{P}}_{s}\left(M\right)\leq \underset{-\frac{1}{2}\leq \theta \leq 0}{\inf}M^{\frac{\theta }{1+\theta }}e^{-\frac{\theta }{1+\theta }H_{1+\theta }^{↑}\left(X|Y\right)}. \end{array}$$

Although Lemma 14 is known to be tight in the large deviation regime for i.i.d. sources, $H_{1+\theta }^{↑}\left(X|Y\right)$ for Markov chains can only be evaluated under the strongly non-hidden assumption. For this reason, even though the following bound is looser than Lemma 14, it is useful to have another bound in terms of $H_{1+\theta }^{↓}\left(X|Y\right)$, which can be evaluated for Markov chains under the non-hidden assumption.

**Lemma** **15.**
*The following bound holds:*
(119)$$\begin{array}{c} {\bar{P}}_{s}\left(M\right)\leq \underset{-1\leq \theta \leq 0}{\inf}M^{\theta }e^{-\theta H_{1+\theta }^{↓}\left(X|Y\right)}. \end{array}$$


**Proof.** To derive this bound, we change the variable in (Equation 118) as $\theta =\frac{\theta^{\prime }}{1-\theta^{\prime }}$. Then, $-1\leq \theta^{\prime }\leq 0$, and we have:
$$\begin{array}{c} M^{\theta^{\prime }}e^{-\theta^{\prime }H_{\frac{1}{1-\theta^{\prime }}}^{↑}\left(X|Y\right)}\leq M^{\theta^{\prime }}e^{-\theta^{\prime }H_{1+\theta^{\prime }}^{↓}\left(X|Y\right)}, \end{array}$$
where we use Lemma A4 in Appendix C. □

For the source coding without side-information, i.e., when *X* has no side-information, we have the following bound, which is tighter than Lemma 14.

**Lemma** **16**((2.39) [65]). *The following bound holds:*
(120)$$\begin{array}{c} P_{s}\left(M\right)\leq \underset{-1<\theta \leq 0}{\inf}M^{\frac{\theta }{1+\theta }}e^{-\frac{\theta }{1+\theta }H_{1+\theta }\left(X\right)}. \end{array}$$

For the converse part, we first have the following bound, which is very close to the operational definition of source coding with side-information.

**Lemma** **17**([66]). *Let ${\left\{\Omega_{y}\right\}}_{y\in Y}$ be a family of subsets $\Omega_{y}\subset X$, and let $\Omega =\cup_{y\in Y}\Omega_{y}\times \left\{y\right\}$. Then, for any $Q_{Y}\in P\left(Y\right)$, the following bound holds:*
(121)$$\begin{array}{c} P_{s}\left(M\right)\geq \min_{\left\{\Omega_{y}\right\}}\left\{P_{XY}\left(\Omega^{c}\right):\sum_{y}Q_{Y}\left(y\right)\left|\Omega_{y}\right|\leq M\right\}. \end{array}$$

Since Lemma 17 is close to the operational definition, it is not easy to evaluate Lemma 17. Thus, we derive another bound by loosening Lemma 17, which is more tractable for evaluation. Slightly weakening Lemma 17, we have the following.

**Lemma** **18**([3,4]). *For any $Q_{Y}\in P\left(Y\right)$, we have (In fact, a special case for $Q_{Y}=P_{Y}$ corresponds to Lemma 7.2.2 of [3]. A bound that involves $Q_{Y}$ was introduced in [4] for channel coding, and it can be regarded as a source coding counterpart of that result.):*
(122)$$\begin{array}{c} P_{s}\left(M\right)\geq \underset{\gamma \geq 0}{\sup}\left[P_{XY}\left\{\log\frac{Q_{Y}\left(y\right)}{P_{XY}\left(x,y\right)}>\gamma \right\}-\frac{M}{e^{\gamma }}\right]. \end{array}$$

By using the change-of-measure argument, we also obtain the following converse bound.

**Theorem** **5.**
*For any $Q_{Y}\in P\left(Y\right)$, we have:*
(123)−logPs(M)≤infs>0θ˜∈R,ϑ≥0[(1+s)θ˜H1+θ˜(PXY|QY)−H1+(1+s)θ˜(PXY|QY)
(124)−(1+s)log1−2e−−ϑR+(θ˜+ϑ(1+θ˜))Hθ˜+ϑ(1+θ˜)(PXY|QY)−(1+ϑ)θ˜H1+θ˜(PXY|QY)1+ϑ]/s≤infs>0−1<θ˜<θ(a(R))[(1+s)θ˜H1+θ˜(PXY|QY)−H1+(1+s)θ˜(PXY|QY)
(125)$$\begin{array}{c} -\left(1+s\right)\log\left(1-2e^{\left(\theta \left(a\left(R\right)\right)-\tilde{\theta }\right)a\left(R\right)-\theta \left(a\left(R\right)\right)H_{1+\theta (a(R))}\left(P_{XY}|Q_{Y}\right)+\tilde{\theta }H_{1+\tilde{\theta }}\left(P_{XY}|Q_{Y}\right)}\right)]/s, \end{array}$$
*where R=logM, and $\theta \left(a\right)=\theta^{Q}\left(a\right)$ and $a\left(R\right)=a^{Q}\left(R\right)$ are the inverse functions defined in (Equation 29) and (Equation 32), respectively.*


**Proof.** See Appendix K. □

In particular, by taking $Q_{Y}=P_{Y}^{\left(1+\theta (a(R))\right)}$ in Theorem 5, we have the following.

**Corollary** **1.**
*We have:*
(126)−logPs(M)≤infs>0−1<θ˜<θ(a(R))[(1+s)θ˜H1+θ˜,1+θ(a(R))(X|Y)−H1+(1+s)θ˜,1+θ(a(R))(X|Y)
(127)$$\begin{array}{c} -\left(1+s\right)\log\left(1-2e^{\left(\theta \left(a\left(R\right)\right)-\tilde{\theta }\right)a\left(R\right)-\theta \left(a\left(R\right)\right)H_{1+\theta (a(R))}^{↑}\left(X|Y\right)+\tilde{\theta }H_{1+\tilde{\theta },1+\theta \left(a\left(R\right)\right)}\left(X|Y\right)}\right)]/s, \end{array}$$

*where $\theta \left(a\right)=\theta^{↑}\left(a\right)$ and $a\left(R\right)=a^{↑}\left(R\right)$ are the inverse functions defined in (Equation 35) and (Equation 36).*


**Remark** **5.**
*Here, we discuss the possibility for extension to the continuous case. As explained in Remark 1, we can define the information quantities for the case when Y is continuous, but X is a discrete finite set. The discussions in this subsection still hold even in this continuous case. In particular, in the n-i.i.d. extension case with this continuous setting, Lemma 14 and Corollary 1 hold when the information measures are replaced by n times the single-shot information measures.*


### 3.3. Finite-Length Bounds for Markov Source

In this subsection, we derive several finite-length bounds for the Markov source with a computable form. Unfortunately, it is not easy to evaluate how tight those bounds are only with their formula. Their tightness will be discussed by considering the asymptotic limit in the remaining subsections of this section. Since we assume the irreducibility for the transition matrix describing the Markov chain, the following bound holds with any initial distribution.

To derive a lower bound on $-\log{\bar{P}}_{s}\left(M_{n}\right)$ in terms of the Rényi entropy of the transition matrix, we substitute the formula for the Rényi entropy given in Lemma 10 into Lemma 15. Then, we can derive the following achievability bound.

**Theorem** **6**(Direct, Ass. 1). *Suppose that transition matrix W satisfies Assumption 1. Let $R:=\frac{1}{n}\log M_{n}$. Then, for every n≥1, we have:*
(128)$$\begin{array}{c} -\log{\bar{P}}_{s}^{\left(n\right)}\left(M_{n}\right)\geq \underset{-1\leq \theta \leq 0}{\sup}\left[-\theta nR+\left(n-1\right)\theta H_{1+\theta }^{↓,W}\left(X|Y\right)+\underline{\delta }\left(\theta \right)\right], \end{array}$$
*where $\underline{\delta }\left(\theta \right)$ is given by (91).*


For the source coding without side-information, from Lemma 16 and a special case of Lemma 10, we have the following achievability bound.

**Theorem** **7**(Direct, no-side-information). *Let $R:=\frac{1}{n}\log M_{n}$. Then, for every n≥1, we have:*
(129)$$\begin{array}{c} -\log P_{e}^{\left(n\right)}\left(M_{n}\right)\geq \underset{-1<\theta \leq 0}{\sup}\frac{-n\theta R+\left(n-1\right)\theta H_{1+\theta }^{W}\left(X\right)+\underline{\delta }\left(\theta \right)}{1+\theta }. \end{array}$$

To derive an upper bound on $-\log P_{s}\left(M_{n}\right)$ in terms of the Rényi entropy of transition matrix, we substitute the formula for the Rényi entropy given in Lemma 10 for Theorem 5. Then, we have the following converse bound.

**Theorem** **8**(Converse, Ass. 1). *Suppose that transition matrix W satisfies Assumption 1. Let $R:=\frac{1}{n}\log M_{n}$. For any $H^{W}\left(X|Y\right)<R<H_{0}^{↓,W}\left(X|Y\right)$, we have:*
(130)−logPs(n)(Mn)≤infs>0−1<θ˜<θ(a(R))[(n−1)(1+s)θ˜H1+θ˜↓,W(X|Y)−H1+(1+s)θ˜↓,W(X|Y)+δ1
(131)$$\begin{array}{c} -\left(1+s\right)\log\left(1-2e^{\left(n-1\right)\left[\left(\theta^{W}\left(a^{W}\left(R\right)\right)-\tilde{\theta }\right)a^{W}\left(R\right)-\theta^{W}\left(a^{W}\left(R\right)\right)H_{1+\theta^{W}\left(a^{W}\left(R\right)\right)}^{↓,W}\left(X|Y\right)+\tilde{\theta }H_{1+\tilde{\theta }}^{↓,W}\left(X|Y\right)\right]+\delta_{2}}\right)]/s, \end{array}$$
*where $\theta \left(a\right)=\theta^{↓}\left(a\right)$ and $a\left(R\right)=a^{↓}\left(R\right)$ are the inverse functions defined by (Equation 67) and (Equation 70), respectively,*
(132)$$\begin{array}{ccc} \delta_{1} & := & \left(1+s\right)\bar{\delta }\left(\tilde{\theta }\right)-\underline{\delta }\left(\left(1+s\right)\tilde{\theta }\right), \end{array}$$
(133)$$\begin{array}{ccc} \delta_{2} & := & \frac{\left(\theta^{W}\left(a^{W}\left(R\right)\right)-\tilde{\theta }\right)R-\left(1+\tilde{\theta }\right)\underline{\delta }\left(\theta^{W}\left(a^{W}\left(R\right)\right)\right)+\left(1+\theta^{W}\left(a^{W}\left(R\right)\right)\right)\bar{\delta }\left(\tilde{\theta }\right)}{1+\theta^{W}\left(a^{W}\left(R\right)\right)}, \end{array}$$
*and $\bar{\delta }\left(\cdot \right)$ and $\underline{\delta }\left(\cdot \right)$ are given by (Equation 90) and (91), respectively.*


**Proof.** We first use (Equation 124) of Theorem 5 for $Q_{Y^{n}}=P_{Y^{n}}$ and Lemma 10. Then, we restrict the range of $\tilde{\theta }$ as $-1<\tilde{\theta }<\theta^{W}\left(a^{W}\left(R\right)\right)$ and set $\vartheta =\frac{\theta^{W}\left(a^{W}\left(R\right)\right)-\tilde{\theta }}{1+\tilde{\theta }}$. Then, we have the assertion of the theorem. □

Next, we derive tighter bounds under Assumption 2. To derive a lower bound on $-\log{\bar{P}}_{s}\left(M_{n}\right)$ in terms of the Rényi entropy of the transition matrix, we substitute the formula for the Rényi entropy in Lemma 11 for Lemma 14. Then, we have the following achievability bound.

**Theorem** **9**(Direct, Ass. 2). *Suppose that transition matrix W satisfies Assumption 2. Let $R:=\frac{1}{n}\log M_{n}$. Then, we have:*
(134)$$\begin{array}{c} -\log{\bar{P}}_{s}^{\left(n\right)}\left(M_{n}\right)\geq \underset{-\frac{1}{2}\leq \theta \leq 0}{\sup}\frac{-\theta nR+\left(n-1\right)\theta H_{1+\theta }^{↑,W}\left(X|Y\right)}{1+\theta }+\underline{\xi }\left(\theta \right), \end{array}$$
*where $\underline{\xi }\left(\theta \right)$ is given by (98).*


Finally, to derive an upper bound on $-\log P_{s}\left(M_{n}\right)$ in terms of the Rényi entropy for the transition matrix, we substitute the formula for the Rényi entropy in Lemma 12 for Theorem 5 for $Q_{Y^{n}}=P_{Y^{n}}^{\left(1+\theta^{W}\left(a^{W}\left(R\right)\right)\right)}$. Then, we can derive the following converse bound.

**Theorem** **10**(Converse, Ass. 2). *Suppose that transition matrix W satisfies Assumption 2. Let $R:=\frac{1}{n}\log M_{n}$. For any $H^{W}\left(X|Y\right)<R<H_{0}^{↑,W}\left(X|Y\right)$, we have:*
(135)−logPs(n)(Mn)≤infs>0−1<θ˜<θW(aW(R))[(n−1)(1+s)θ˜H1+θ˜,1+θW(aW(R))W(X|Y)−H1+(1+s)θ˜,1+θW(aW(R))W(X|Y)+δ1
(136)$$\begin{array}{cc} \; & -\left(1+s\right)\log\left(1-2e^{\left(n-1\right)\left[\left(\theta^{W}\left(a^{W}\left(R\right)\right)-\tilde{\theta }\right)a^{W}\left(R\right)-\theta^{W}\left(a^{W}\left(R\right)\right)H_{1+\theta^{W}\left(a^{W}\left(R\right)\right)}^{↑,W}\left(X|Y\right)+\tilde{\theta }H_{1+\tilde{\theta },1+\theta^{W}\left(a^{W}\left(R\right)\right)}^{W}\left(X|Y\right)\right]+\delta_{2}}\right)]/s, \end{array}$$
*where $\theta^{W}\left(a\right)=\theta^{↑,W}\left(a\right)$ and $a^{W}\left(R\right)=a^{↑,W}\left(R\right)$ are the inverse functions defined by (Equation 71) and (Equation 73), respectively,*
(137)$$\begin{array}{cc} \delta_{1}:= & \left(1+s\right)\bar{\zeta }\left(\tilde{\theta },\theta^{W}\left(a^{W}\left(R\right)\right)\right)-\underline{\zeta }\left(\left(1+s\right)\tilde{\theta },\theta^{W}\left(a^{W}\left(R\right)\right)\right), \end{array}$$
(138)$$\begin{array}{cc} \delta_{2}:= & \frac{\left(\theta^{W}\left(a^{W}\left(R\right)\right)-\tilde{\theta }\right)R-\left(1+\tilde{\theta }\right)\underline{\zeta }\left(\theta^{W}\left(a^{W}\left(R\right)\right),\theta^{W}\left(a^{W}\left(R\right)\right)\right)+\left(1+\theta^{W}\left(a^{W}\left(R\right)\right)\right)\bar{\zeta }\left(\tilde{\theta },\theta^{W}\left(a^{W}\left(R\right)\right)\right)}{1+\theta^{W}\left(a^{W}\left(R\right)\right)}, \end{array}$$
*and $\bar{\zeta }\left(\cdot ,\cdot \right)$ and $\underline{\zeta }\left(\cdot ,\cdot \right)$ are given by (Equation 102)–(105).*


**Proof.** We first use (Equation 124) of Theorem 5 for $Q_{Y^{n}}=P_{Y^{n}}^{\left(1+\theta^{W}\left(a^{W}\left(R\right)\right)\right)}$ and Lemma 12. Then, we restrict the range of $\tilde{\theta }$ as $-1<\tilde{\theta }<\theta^{W}\left(a^{W}\left(R\right)\right)$ and set $\vartheta =\frac{\theta^{W}\left(a^{W}\left(R\right)\right)-\tilde{\theta }}{1+\tilde{\theta }}$. Then, we have the assertion of the theorem. □

### 3.4. Second-Order

By applying the central limit theorem to Lemma 13 (cf. [67] (Theorem 27.4, Example 27.6)) and Lemma 18 for $Q_{Y}=P_{Y}$ and by using Theorem 2, we have the following.

**Theorem** **11.**
*Suppose that transition matrix W on X×Y satisfies Assumption 1. For arbitrary ε∈(0,1), we have:*
(139)$$\begin{array}{c} \log M\left(n,\varepsilon \right)=\log\bar{M}\left(n,\varepsilon \right)+o\left(\sqrt{n}\right)=nH^{W}\left(X|Y\right)+\sqrt{V^{W}\left(X|Y\right)}\sqrt{n}\Phi \left(1-\varepsilon \right)+o\left(\sqrt{n}\right). \end{array}$$


**Proof.** The central limit theorem for the Markov process cf. [67] (Theorem 27.4, Example 27.6) guarantees that the random variable $\left(-\log P_{X^{n}|Y^{n}}\left(X^{n}|Y^{n}\right)-nH^{W}\left(X|Y\right)\right)/\sqrt{n}$ asymptotically obeys the normal distribution with average zero and variance $V^{W}\left(X|Y\right)$, where we use Theorem 2 to show that the limit of the variance is given by $V^{W}\left(X|Y\right)$. Let $R=\sqrt{V^{W}\left(X|Y\right)}\Phi^{-1}\left(1-\varepsilon \right)$. Substituting $M=e^{nH^{W}\left(X|Y\right)+\sqrt{n}R}$ and $\gamma =nH^{W}\left(X|Y\right)+\sqrt{n}R-n^{\frac{1}{4}}$ in Lemma 13, we have:
(140)$$\begin{array}{c} \lim_{n\to \infty }{\bar{P}}_{s}^{\left(n\right)}\left(e^{nH^{W}\left(X|Y\right)+\sqrt{n}R}\right)\leq \varepsilon . \end{array}$$ □On the other hand, substituting $M=e^{nH^{W}\left(X|Y\right)+\sqrt{n}R}$ and $\gamma =nH^{W}\left(X|Y\right)+\sqrt{n}R+n^{\frac{1}{4}}$ in Lemma 18 for $Q_{Y}=P_{Y}$, we have:
(141)$$\begin{array}{c} \lim_{n\to \infty }P_{s}^{\left(n\right)}\left(e^{nH^{W}\left(X|Y\right)+\sqrt{n}R}\right)\geq \varepsilon . \end{array}$$Combining (Equation 140) and (Equation 141), we have the statement of the theorem.

From the above theorem, the (first-order) compression limit of source coding with side-information for a Markov source under Assumption 1 is given by (although the compression limit of source coding with side-information for a Markov chain is known more generally [68], we need Assumption 1 to get a single-letter characterization):(142)$$\begin{array}{ccc} \lim_{n\to \infty }\frac{1}{n}\log M\left(n,\varepsilon \right) & = & \lim_{n\to \infty }\frac{1}{n}\log\bar{M}\left(n,\varepsilon \right) \end{array}$$(143)$$\begin{array}{cc} = & H^{W}\left(X|Y\right) \end{array}$$
for any ε∈(0,1). In the next subsections, we consider the asymptotic behavior of the error probability when the rate is larger than the compression limit $H^{W}\left(X|Y\right)$ in the moderate deviation regime and the large deviation regime, respectively.

### 3.5. Moderate Deviation

From Theorems 6 and 8, we have the following.

**Theorem** **12.**
*Suppose that transition matrix W satisfies Assumption 1. For arbitrary t∈(0,1/2) and δ>0, we have:*
(144)$$\begin{array}{ccc} \lim_{n\to \infty }-\frac{1}{n^{1-2t}}\log P_{s}^{\left(n\right)}\left(e^{nH^{W}\left(X|Y\right)+n^{1-t}\delta }\right) & = & \lim_{n\to \infty }-\frac{1}{n^{1-2t}}\log{\bar{P}}_{s}^{\left(n\right)}\left(e^{nH^{W}\left(X|Y\right)+n^{1-t}\delta }\right) \end{array}$$
(145)$$\begin{array}{cc} = & \frac{\delta^{2}}{2V^{W}\left(X|Y\right)}. \end{array}$$


**Proof.** We apply Theorems 6 and 8 to the case with $R=H^{W}\left(X|Y\right)+n^{-t}\delta$, i.e., $\theta \left(a\left(R\right)\right)=-n^{-1}\frac{\delta }{V^{W}\left(X|Y\right)}+o\left(n^{-t}\right)$. For the achievability part, from (Equation 88) and Theorem 6, we have:
(146)$$\begin{array}{cc} -\log P_{s}^{\left(n\right)}\left(M_{n}\right)\; & \geq \underset{-1\leq \theta \leq 0}{\sup}\left[-\theta nR+\left(n-1\right)\theta H_{1+\theta }^{↓,W}\left(X|Y\right)\right]+\underset{-1\leq \theta \leq 0}{\inf}\underline{\delta }\left(\theta \right) \end{array}$$
(147)$$\begin{array}{cc} \; & \geq n^{1-2t}\frac{\delta^{2}}{2V^{W}\left(X|Y\right)}+o\left(n^{1-2t}\right). \end{array}$$To prove the converse part, we fix arbitrary s>0 and choose $\tilde{\theta }$ to be $-n^{-t}\frac{\delta }{V^{W}\left(X|Y\right)}+n^{-2t}$. Then, Theorem 8 implies that:
(148)$$\begin{array}{cc} \underset{n\to \infty }{lim sup}-\frac{1}{n^{1-2t}}\log P_{s}\left(M_{n}\right)\; & \leq \underset{n\to \infty }{lim sup}n^{2t}\frac{1+s}{s}\tilde{\theta }\left\{H_{1+\tilde{\theta }}^{↓,W}\left(X|Y\right)-H_{1+\left(1+s\right)\tilde{\theta }}^{↓,W}\left(X|Y\right)\right\} \end{array}$$
(149)$$\begin{array}{cc} \; & =\underset{n\to \infty }{lim sup}n^{2t}\frac{1+s}{s}s{\tilde{\theta }}^{2}\frac{dH_{1+\theta }^{↓,W}\left(X|Y\right)}{d\theta }{|}_{\theta =\tilde{\theta }} \end{array}$$
(150)$$\begin{array}{cc} \; & =\left(1+s\right)\frac{\delta^{2}}{2V^{W}\left(X|Y\right)}. \end{array}$$ □

**Remark** **6.**
*In the literature [13,69], the moderate deviation results are stated for $\epsilon_{n}$ such that $\epsilon_{n}\to 0$ and $n\epsilon_{n}^{2}\to \infty$ instead of $n^{-t}$ for t∈(0,1/2). Although the former is slightly more general than the latter, we employ the latter formulation in Theorem 12 since the order of convergence is clearer. In fact, $n^{-t}$ in Theorem 12 can be replaced by general $\epsilon_{n}$ without modifying the argument of the proof.*


### 3.6. Large Deviation

From Theorems 6 and 8, we have the following.

**Theorem** **13.**
*Suppose that transition matrix W satisfies Assumption 1. For $H^{W}\left(X|Y\right)<R$, we have:*
(151)$$\begin{array}{c} \underset{n\to \infty }{lim inf}-\frac{1}{n}\log{\bar{P}}_{s}^{\left(n\right)}\left(e^{nR}\right)\geq \underset{-1\leq \theta \leq 0}{\sup}\left[-\theta R+\theta H_{1+\theta }^{↓,W}\left(X|Y\right)\right]. \end{array}$$

*On the other hand, for $H^{W}\left(X|Y\right)<R<H_{0}^{↓,W}\left(X|Y\right)$, we have:*
(152)$$\begin{array}{ccc} \underset{n\to \infty }{lim sup}-\frac{1}{n}\log P_{s}^{\left(n\right)}\left(e^{nR}\right) & \leq & -\theta \left(a\left(R\right)\right)a\left(R\right)+\theta \left(a\left(R\right)\right)H_{1+\theta (a(R))}^{↓,W}\left(X|Y\right) \end{array}$$
(153)$$\begin{array}{cc} = & \underset{-1<\theta \leq 0}{\sup}\frac{-\theta R+\theta H_{1+\theta }^{↓,W}\left(X|Y\right)}{1+\theta }. \end{array}$$


**Proof.** The achievability bound (Equation 151) follows from Theorem 6. The converse part (Equation 152) is proven from Theorem 8 as follows. We first fix s>0 and $-1<\tilde{\theta }<\theta \left(a\left(R\right)\right)$. Then, Theorem 8 implies:
(154)$$\begin{array}{c} \underset{n\to \infty }{lim sup}-\frac{1}{n}\log P_{s}^{\left(n\right)}\left(e^{nR}\right)\leq \frac{1+s}{s}\tilde{\theta }\left\{H_{1+\tilde{\theta }}^{↓,W}\left(X|Y\right)-H_{1+\left(1+s\right)\tilde{\theta }}^{↓,W}\left(X|Y\right)\right\}. \end{array}$$By taking the limit s→0 and $\tilde{\theta }\to \theta \left(a\left(R\right)\right)$, we have:
(155)$$\begin{array}{cc} \; & \frac{1+s}{s}\tilde{\theta }\left\{H_{1+\tilde{\theta }}^{↓,W}\left(X|Y\right)-H_{1+\left(1+s\right)\tilde{\theta }}^{↓,W}\left(X|Y\right)\right\} \end{array}$$
(156)$$\begin{array}{cc} \; & =\frac{1}{s}\left(\tilde{\theta }H_{1+\tilde{\theta }}^{↓,W}\left(X|Y\right)-\left(1+s\right)\tilde{\theta }H_{1+\left(1+s\right)\tilde{\theta }}^{↓,W}\left(X|Y\right)\right)+\tilde{\theta }H_{1+\tilde{\theta }}^{↓,W}\left(X|Y\right) \end{array}$$
(157)$$\begin{array}{cc} \; & \to -\tilde{\theta }\frac{d[\theta H_{1+\theta }^{↓,W}\left(X|Y\right)]}{d\theta }{|}_{\theta =\tilde{\theta }}+\tilde{\theta }H_{1+\tilde{\theta }}^{↓,W}\left(X|Y\right)\;\;\;\left(ass\to 0\right) \end{array}$$
(158)$$\begin{array}{cc} \; & \to -\theta \left(a\left(R\right)\right)\frac{d[\theta H_{1+\theta }^{↓,W}\left(X|Y\right)]}{d\theta }{|}_{\theta =\theta (a(R))}+\theta \left(a\left(R\right)\right)H_{1+\theta (a(R))}^{↓,W}\left(X|Y\right)\;\;\;\left(as\tilde{\theta }\to \theta \left(a\left(R\right)\right)\right) \end{array}$$
(159)$$\begin{array}{cc} \; & =-\theta \left(a\left(R\right)\right)a\left(R\right)+\theta \left(a\left(R\right)\right)H_{1+\theta (a(R))}^{↓,W}\left(X|Y\right). \end{array}$$Thus, (Equation 152) is proven. The alternative expression (153) is derived via Lemma 9. □

Under Assumption 2, from Theorems 9 and 10, we have the following tighter bound.

**Theorem** **14.**
*Suppose that transition matrix W satisfies Assumption 2. For $H^{W}\left(X|Y\right)<R$, we have:*
(160)$$\begin{array}{c} \underset{n\to \infty }{lim inf}-\frac{1}{n}\log{\bar{P}}_{s}^{\left(n\right)}\left(e^{nR}\right)\geq \underset{-\frac{1}{2}\leq \theta \leq 0}{\sup}\frac{-\theta R+\theta H_{1+\theta }^{↑,W}\left(X|Y\right)}{1+\theta }. \end{array}$$

*On the other hand, for $H^{W}\left(X|Y\right)<R<H_{0}^{↑,W}\left(X|Y\right)$, we have:*
(161)$$\begin{array}{ccc} \underset{n\to \infty }{lim sup}-\frac{1}{n}\log P_{s}^{\left(n\right)}\left(e^{nR}\right) & \leq & -\theta \left(a\left(R\right)\right)a\left(R\right)+\theta \left(a\left(R\right)\right)H_{1+\theta (a(R))}^{↑,W}\left(X|Y\right) \end{array}$$
(162)$$\begin{array}{cc} = & \underset{-1<\theta \leq 0}{\sup}\frac{-\theta R+\theta H_{1+\theta }^{↑,W}\left(X|Y\right)}{1+\theta }. \end{array}$$


**Proof.** The achievability bound (Equation 160) follows from Theorem 9. The converse part (Equation 161) is proven from Theorem 10 as follows. We first fix s>0 and $-1<\tilde{\theta }<\theta \left(a\left(R\right)\right)$. Then, Theorem 10 implies:
(163)$$\begin{array}{c} \underset{n\to \infty }{lim sup}-\frac{1}{n}\log P_{s}^{\left(n\right)}\left(e^{nR}\right)\leq \frac{1+s}{s}\tilde{\theta }\left\{H_{1+\tilde{\theta },1+\theta \left(a\left(R\right)\right)}^{W}\left(X|Y\right)-H_{1+\left(1+s\right)\tilde{\theta },1+\theta \left(a\left(R\right)\right)}^{W}\left(X|Y\right)\right\}. \end{array}$$By taking the limit s→0 and $\tilde{\theta }\to \theta \left(a\left(R\right)\right)$, we have:
(164)$$\begin{array}{cc} \; & \frac{1+s}{s}\tilde{\theta }\left\{H_{1+\tilde{\theta },1+\theta \left(a\left(R\right)\right)}^{W}\left(X|Y\right)-H_{1+\left(1+s\right)\tilde{\theta },1+\theta \left(a\left(R\right)\right)}^{W}\left(X|Y\right)\right\} \end{array}$$
(165)$$\begin{array}{cc} \; & =\frac{1}{s}\left(\tilde{\theta }H_{1+\tilde{\theta },1+\theta \left(a\left(R\right)\right)}^{W}\left(X|Y\right)-\left(1+s\right)\tilde{\theta }H_{1+\left(1+s\right)\tilde{\theta },1+\theta \left(a\left(R\right)\right)}^{W}\left(X|Y\right)\right)+\tilde{\theta }H_{1+\tilde{\theta },1+\theta \left(a\left(R\right)\right)}^{W}\left(X|Y\right) \end{array}$$
(166)$$\begin{array}{cc} \; & \to -\tilde{\theta }\frac{d[\theta H_{1+\theta ,1+\theta (a(R))}^{W}\left(X|Y\right)]}{d\theta }{|}_{\theta =\tilde{\theta }}+\tilde{\theta }H_{1+\tilde{\theta },1+\theta \left(a\left(R\right)\right)}^{W}\left(X|Y\right)\;\;\;\left(ass\to 0\right) \end{array}$$
(167)$$\begin{array}{cc} \; & \to -\theta \left(a\left(R\right)\right)\frac{d[\theta H_{1+\theta ,1+\theta (a(R))}^{W}\left(X|Y\right)]}{d\theta }{|}_{\theta =\theta (a(R))}+\theta \left(a\left(R\right)\right)H_{1+\theta (a(R))}^{↑,W}\left(X|Y\right)\;\;\;\left(as\tilde{\theta }\to \theta \left(a\left(R\right)\right)\right) \end{array}$$
(168)$$\begin{array}{cc} \; & =-\theta \left(a\left(R\right)\right)a\left(R\right)+\theta \left(a\left(R\right)\right)H_{1+\theta (a(R))}^{↑,W}\left(X|Y\right). \end{array}$$Thus, (Equation 161) is proven. The alternative expression (162) is derived via Lemma 9.□

**Remark** **7.**
*For $R\leq R_{\mathrm{cr}}$, where (cf. (Equation 72) for the definition of R(a)):*
(169)$$\begin{array}{c} R_{\mathrm{cr}}:=R\left(\frac{d[\theta H_{1+\theta }^{↑,W}\left(X|Y\right)]}{d\theta }{|}_{\theta =-\frac{1}{2}}\right) \end{array}$$
*is the critical rate, the left-hand side of (Equation 76) in Lemma 9 is attained by parameters in the range −1/2≤θ≤0. Thus, the lower bound in (Equation 160) is rewritten as:*
(170)$$\begin{array}{c} \underset{-\frac{1}{2}\leq \theta \leq 0}{\sup}\frac{-\theta R+\theta H_{1+\theta }^{↑,W}\left(X|Y\right)}{1+\theta }=-\theta \left(a\left(R\right)\right)a\left(R\right)+\theta \left(a\left(R\right)\right)H_{1+\theta (a(R))}^{↑,W}\left(X|Y\right). \end{array}$$

*Thus, the lower bound and the upper bounds coincide up to the critical rate.*


**Remark** **8.**
*For the source coding without side-information, by taking the limit of Theorem 7, we have:*
(171)$$\begin{array}{c} \underset{n\to \infty }{lim inf}-\frac{1}{n}\log{\bar{P}}_{s}^{\left(n\right)}\left(e^{nR}\right)\geq \underset{-1\leq \theta \leq 0}{\sup}\frac{-\theta R+\theta H_{1+\theta }^{W}\left(X\right)}{1+\theta }. \end{array}$$

*On the other hand, as a special case of (Equation 152) without side-information, we have:*
(172)$$\begin{array}{ccc} \underset{n\to \infty }{lim sup}-\frac{1}{n}\log P_{s}^{\left(n\right)}\left(e^{nR}\right) & \leq & \underset{-1<\theta \leq 0}{\sup}\frac{-\theta R+\theta H_{1+\theta }^{W}\left(X\right)}{1+\theta } \end{array}$$
*for $H^{W}\left(X\right)<R<H_{0}^{W}\left(X\right)$. Thus, we can recover the results in [40,41] by our approach.*


### 3.7. Numerical Example

In this section, to demonstrate the advantage of our finite-length bound, we numerically evaluate the achievability bound in Theorem 7 and a special case of the converse bound in Theorem 8 for the source coding without side-information. Thanks to the aspect (A2), our numerical calculation shows that our upper finite-length bounds are very close to our lower finite-length bounds when the size *n* is sufficiently large. Thanks to the aspect (A1), we could calculate both bounds with the huge size $n=1\times 10^{5}$ because the calculation complexity behaves as O(1).

We consider a binary transition matrix *W* given by Figure 2, i.e.,
(173)$$\begin{array}{c} W=\left[\begin{array}{cc} 1-p & q\\ p & 1-q \end{array}\right]. \end{array}$$

In this case, the stationary distribution is:(174)$$\begin{array}{ccc} \tilde{P}\left(0\right) & = & \frac{q}{p+q}, \end{array}$$(175)$$\begin{array}{ccc} \tilde{P}\left(1\right) & = & \frac{p}{p+q}. \end{array}$$

The entropy is:(176)$$\begin{array}{c} H^{W}\left(X\right)=\frac{q}{p+q}h\left(p\right)+\frac{p}{p+q}h\left(q\right), \end{array}$$
where h(·) is the binary entropy function. The tilted transition matrix is:(177)$$\begin{array}{c} W_{\theta }=\left[\begin{array}{cc} {\left(1-p\right)}^{1+\theta } & q^{1+\theta }\\ p^{1+\theta } & {\left(1-q\right)}^{1+\theta } \end{array}\right]. \end{array}$$

The Perron–Frobenius eigenvalue is:(178)$$\begin{array}{c} \lambda_{\theta }=\frac{{\left(1-p\right)}^{1+\theta }+{\left(1-q\right)}^{1+\theta }+\sqrt{{\left\{{\left(1-p\right)}^{1+\theta }-{\left(1-q\right)}^{1+\theta }\right\}}^{2}+4p^{1+\theta }q^{1+\theta }}}{2} \end{array}$$
and its normalized eigenvector is:(179)$$\begin{array}{ccc} {\tilde{P}}_{\theta }\left(0\right) & = & \frac{q^{1+\theta }}{\lambda_{\theta }-{\left(1-p\right)}^{1+\theta }+q^{1+\theta }}, \end{array}$$(180)$$\begin{array}{ccc} {\tilde{P}}_{\theta }\left(1\right) & = & \frac{\lambda_{\theta }-{\left(1-p\right)}^{1+\theta }}{\lambda_{\theta }-{\left(1-p\right)}^{1+\theta }+q^{1+\theta }}. \end{array}$$

The normalized eigenvector of $W_{\rho }^{T}$ is also given by:(181)$$\begin{array}{ccc} {\hat{P}}_{\theta }\left(0\right) & = & \frac{p^{1+\theta }}{\lambda_{\theta }-{\left(1-p\right)}^{1+\theta }+p^{1+\theta }}, \end{array}$$(182)$$\begin{array}{ccc} {\hat{P}}_{\theta }\left(1\right) & = & \frac{\lambda_{\theta }-{\left(1-p\right)}^{1+\theta }}{\lambda_{\theta }-{\left(1-p\right)}^{1+\theta }+p^{1+\theta }}. \end{array}$$

From these calculations, we can evaluate the bounds in Theorems 7 and 8. For p=0.1, q=0.2, the bounds are plotted in Figure 3 for fixed error probability $\varepsilon =10^{-3}$. Although there is a gap between the achievability bound and the converse bound for rather small *n*, the gap is less than approximately 5% of the entropy rate for *n* larger than 10,000. We also plot the bounds in Figure 4 for fixed block length n= 10,000 and varying ε. The gap between the achievability bound and the converse bound remains approximately 5% of the entropy rate even for ε as small as $10^{-10}$.

The gap between the achievability bound and the converse bound in Figure 3 is rather large compared to a similar numerical experiment conducted in [1]. One reason for the gap is that our bounds are exponential-type bounds. For instance, when the source is i.i.d., the achievability bound essentially reduces to the so-called Gallager bound [64]. However, an advantage of our bounds is that the computational complexity does not depend on the blocklength. The computational complexities of the bounds plotted in [1] depend the blocklength, and numerical computation of those bounds for Markov sources seems to be difficult.

When p=q, an alternative approach to derive tighter bounds is to consider encoding of the Markov transition, i.e., $1[X_{i}=X_{i+1}]$, instead of the source itself (cf. [45] (Example 4)). Then, the analysis can be reduced to i.i.d. case. However, such an approach is possible only when p=q.

### 3.8. Summary of the Results

The obtained results in this section are summarized in Table 2. The check marks 🗸 indicate that the tight asymptotic bounds (large deviation, moderate deviation, and second-order) can be obtained from those bounds. The marks 🗸${}^{*}$ indicate that the large deviation bound can be derived up to the critical rate. The computational complexity “Tail” indicates that the computational complexities of those bounds depend on the computational complexities of tail probabilities. It should be noted that Theorem 8 is derived from a special case ($Q_{Y}=P_{Y}$) of Theorem 5. The asymptotically optimal choice is $Q_{Y}=P_{Y}^{\left(1+\theta \right)}$, which corresponds to Corollary 1. Under Assumption 1, we can derive the bound of the Markov case only for that special choice of $Q_{Y}$, while under Assumption 2, we can derive the bound of the Markov case for the optimal choice of $Q_{Y}$.

## 4. Channel Coding

In this section, we investigate the channel coding with a conditional additive channel. The first part of this section discusses the general properties of the channel coding with a conditional additive channel. The second part of this section discusses the properties of the channel coding when the conditional additive noise of the channel is Markov. The first part starts with showing the problem setting in Section 4.1 by introducing a conditional additive channel. Section 4.2 gives a canonical method to convert a regular channel to a conditional additive channel. Section 4.3 gives a method to convert a BPSK-AWGN channel to a conditional additive channel. Then, we show some single-shot achievability bounds in Section 4.4 and single-shot converse bounds in Section 4.5.

As the second part, we derive finite-length bounds for the Markov noise channel in Section 4.6. Then, we derive the second-order rate in Section 4.7. In Section 4.8 and Section 4.9, we show the asymptotic characterization for the large deviation regime and the moderate deviation regime by using those finite-length bounds.

### 4.1. Formulation for the Conditional Additive Channel

#### 4.1.1. Single-Shot Case

We first present the problem formulation in the single-shot setting. For a channel $P_{B|A}\left(b|a\right)$ with input alphabet A and output alphabet B, a channel code Ψ=(e,d) consists of one encoder e:{1,…,M}→A and one decoder d:B→{1,…,M}. The average decoding error probability is defined by:(183)$$\begin{array}{c} P_{c}\left[\Psi \right]:=\sum_{m=1}^{M}\frac{1}{M}P_{B|A}\left(\left\{b:d\left(b\right)\neq m\right\}|e\left(m\right)\right). \end{array}$$

For notational convenience, we introduce the error probability under the condition that the message size is *M*:(184)$$\begin{array}{c} P_{c}\left(M\right):=\underset{\Psi }{\inf}P_{c}\left[\Psi \right]. \end{array}$$

Assume that the input alphabet A is the same set as the output alphabet B and they equal an additive group X. When the transition matrix $P_{B|A}\left(b|a\right)$ is given as $P_{X}\left(b-a\right)$ by using a distribution $P_{X}$ on X, the channel is called additive.

To extend the concept of the additive channel, we consider the case when the input alphabet A is an additive group X and the output alphabet B is the product set X×Y. When the transition matrix $P_{B|A}\left(x,y|a\right)$ is given as $P_{XY}\left(x-a,y\right)$ by using a distribution $P_{XY}$ on X×Y, the channel is called conditional additive. In this paper, we are exclusively interested in the conditional additive channel. As explained in Section 4.2, a channel is a conditional additive channel if and only if it is a regular channel in the sense of [31]. When we need to express the underlying distribution of the noise explicitly, we denote the average decoding error probability by $P_{c}\left[\Psi |P_{XY}\right]$.

#### 4.1.2. *n*-Fold Extension

When we consider *n*-fold extension, the channel code is denoted with subscript *n* such as $\Psi_{n}=\left(e_{n},d_{n}\right)$. The error probabilities given in (Equation 183) and (Equation 184) are written with the superscript (n) as $P_{c}^{\left(n\right)}\left[\Psi_{n}\right]$ and $P_{c}^{\left(n\right)}\left(M_{n}\right)$, respectively. Instead of evaluating the error probability $P_{c}^{\left(n\right)}\left(M_{n}\right)$ for given $M_{n}$, we are also interested in evaluating:(185)$$\begin{array}{c} M\left(n,\varepsilon \right):=\sup\left\{M_{n}:P_{c}^{\left(n\right)}\left(M_{n}\right)\leq \varepsilon \right\} \end{array}$$
for given 0≤ε≤1.

When the channel is given as a conditional distribution, the channel is given by:(186)$$\begin{array}{c} P_{B^{n}|A^{n}}\left(x^{n},y^{n}|a^{n}\right)=P_{X^{n}Y^{n}}\left(x^{n}-a^{n},y^{n}\right), \end{array}$$
where $P_{X^{n}Y^{n}}$ is a noise distribution on $X^{n}\times Y^{n}$.

For the code construction, we investigate the linear code. For an (n,k) linear code $C_{n}\subset A^{n}$, there exists a parity check matrix $f_{n}:A^{n}\to A^{n-k}$ such that the kernel of $f_{n}$ is $C_{n}$. That is, given a parity check matrix $f_{n}:A^{n}\to A^{n-k}$, we define the encoder $I_{\mathrm{Ker}(f_{n})}:C_{n}\to A^{n}$ as the imbedding of the kernel $\mathrm{Ker}(f_{n})$. Then, using the decoder $d_{f_{n}}:=\underset{d}{\mathrm{argmin}}P_{c}\left[\left(I_{\mathrm{Ker}(f_{n})},d\right)\right]$, we define $\Psi \left(f_{n}\right)=\left(I_{\mathrm{Ker}(f_{n})},d_{f_{n}}\right)$.

Here, we employ a randomized choice of a parity check matrix. In particular, instead of a two-universal hash function, we focus on linear two-universal hash functions, because the linearity is required in the above relation with source coding. Therefore, denoting the set of linear two-universal hash functions from $A^{n}$ to $A^{n-k}$ by $F_{l}$, we introduce the quantity:(187)$$\begin{array}{c} {\bar{P}}_{c}\left(n,k\right):=\underset{F_{n}\in F_{l}}{\sup}E_{F_{n}}\left[P_{c}^{\left(n\right)}\left[\Psi \left(F_{n}\right)\right]\right]. \end{array}$$

Taking the infimum over all linear codes associated with $F_{n}$ (cf. (Equation 113)), we obviously have:(188)$$\begin{array}{c} P_{c}^{\left(n\right)}{\left(|A\right|}^{k})\leq {\bar{P}}_{c}\left(n,k\right). \end{array}$$

When we consider the error probability for conditionally additive channels, we use notation ${\bar{P}}_{c}\left(n,k|P_{XY}\right)$ so that the underlying distribution of the noise is explicit. We are also interested in characterizing:(189)$$\begin{array}{c} k\left(n,\varepsilon \right):=\sup\left\{k:{\bar{P}}_{c}\left(n,k\right)\leq \varepsilon \right\} \end{array}$$
for given 0≤ε≤1.

### 4.2. Conversion from the Regular Channel to the Conditional Additive Channel

The aim of this subsection is to show the following theorem by presenting the conversion rule between these two types of channels. Then, we see that a binary erasure symmetric channel is an example of a regular channel.

**Theorem** **15.**
*A channel is a regular channel in the sense of [31] if and only if it can be written as a conditional additive channel.*


To show the conversion from a conditional additive channel to a regular channel, we assume that the input alphabet A has an additive group structure. Let $P_{\tilde{X}}$ be a distribution on the output alphabet B. Let $\pi_{a}$ be a representation of the group A on B, and let $G=\{\pi_{a}:a\in A\}$. A regular channel [31] is defined by:(190)$$\begin{array}{c} P_{B|A}\left(b|a\right)=P_{\tilde{X}}\left(\pi_{a}\left(b\right)\right). \end{array}$$

The group action induces orbit:(191)$$\begin{array}{c} \mathrm{Orb}\left(b\right):=\{\pi_{a}\left(b\right):a\in A\}. \end{array}$$

The set of all orbits constitutes a disjoint partition of B. A set of the orbits is denoted by $\bar{B}$, and let $\mathrm{Orb}:B\to \bar{B}$ be the map to the representatives.

**Example** **4**(Binary erasure symmetric channel). *Let A={0,1}, B={0,1,?}, and:*
(192)$$\begin{array}{c} P_{\tilde{X}}\left(b\right)=\left\{\begin{array}{cc} 1-p-p^{\prime } & ifb=0\\ p & ifb=1\\ p^{\prime } & ifb=? \end{array}\right.. \end{array}$$
*Then, let:*
(193)$$\begin{array}{c} \pi_{0}=\left[\begin{array}{ccc} 0 & 1 & ?\\ 0 & 1 & ? \end{array}\right],\;\;\pi_{1}=\left[\begin{array}{ccc} 0 & 1 & ?\\ 1 & 0 & ? \end{array}\right]. \end{array}$$

*The channel defined in this way is a regular channel (see Figure 5). In this case, there are two orbits: {0,1} and {?}.*


Let B=X×Y and $P_{\tilde{X}}=P_{XY}$ for some joint distribution on X×Y. Now, we consider a conditional additive channel, whose transition matrix $P_{B|A}\left(x,y|a\right)$ is given as $P_{XY}\left(x-a,y\right)$. When the group action is given by $\pi_{a}\left(x,y\right)=\left(x-a,y\right)$, the above conditional additive channel is given as a regular channel. In this case, there are |Y| orbits, and the size of each orbit is |X|, respectively. This fact shows that any conditional additive channel is written as a regular channel. That is, it shows the “if” part of Theorem 15.

Conversely, we present the conversion from a regular channel to a conditional additive channel. We first explain the construction for the single-shot channel. For random variable $\tilde{X}∼P_{\tilde{X}}$, let $Y=\bar{B}$ and $Y=\varpi (\tilde{X})$ be the random variable describing the representatives of the orbits. For y=Orb(b) and each orbit Orb(b), we fix an element $0_{y}\in \mathrm{Orb}\left(b\right)$. Then, we define:(194)$$\begin{array}{c} P_{Y}\left(y\right):=P_{\tilde{X}}\left(\mathrm{Orb}\left(b\right)\right),\;P_{X,Y}\left(a,y\right):=\frac{P_{\tilde{X}}\left(\pi_{a}\left(0_{y}\right)\right)}{\left|\{a^{\prime }\in A|\pi_{a}\left(0_{y}\right)=\pi_{a^{\prime }}\left(0_{y}\right)\}\right|}. \end{array}$$

Then, we obtain the virtual channel $P_{X,Y|A}$ as $P_{X,Y|A}\left(x,y|a\right):=P_{X,Y}\left(x-a,y\right)$. Using the conditional distributions $P_{X,Y|B}$ and $P_{B|X,Y}$ as:(195)$$\begin{array}{cc} P_{X,Y|B}\left(a,y|b\right) & =\left\{\begin{array}{cc} \frac{1}{\left|\{a^{\prime }\in A|\pi_{a}\left(0_{y}\right)=\pi_{a^{\prime }}\left(0_{y}\right)\}\right|} & \mathrm{when}\;b=\pi_{a}\left(0_{y}\right)\\ 0 & \mathrm{otherwise}. \end{array}\right. \end{array}$$(196)$$\begin{array}{cc} P_{X,Y|B}\left(a,y|b\right) & =\left\{\begin{array}{cc} 1 & \mathrm{when}\;b=\pi_{a}\left(0_{y}\right)\\ 0 & \mathrm{otherwise}, \end{array}\right. \end{array}$$
we obtain the relations:(197)$$\begin{array}{c} P_{B|A}\left(b|a\right)=\sum_{x,y}P_{B|X,Y}\left(b|x,y\right)P_{X,Y|A}\left(x,y|a\right),\;P_{X,Y|A}\left(x,y|a\right)=\sum_{b}P_{X,Y|B}\left(x,y|b\right)P_{B|A}\left(b|a\right). \end{array}$$

These two equations show that the receiver information of the virtual conditional additive channel $P_{X,Y|A}$ and the receiver information of the regular channel $P_{B|A}$ can be converted into each other. Hence, we can say that a regular channel in the sense of [31] can be written as a conditional additive channel, which shows the “only if” part of Theorem 15.

**Example** **5**(Binary erasure symmetric channel revisited). *We convert the regular channel of Example 4 to a conditional additive channel. Let us label the orbit {0,1} as y=0 and {?} as y=1. Let $0_{0}=0$ and $0_{1}=?$.*
(198)$$\begin{array}{cc} P_{X,Y}\left(x,0\right) & =\left\{\begin{array}{cc} 1-p-p^{\prime } & ifx=0\\ p & ifx=0 \end{array}\right. \end{array}$$
(199)$$\begin{array}{cc} P_{X,Y}\left(x,1\right) & =\frac{p^{\prime }}{2}. \end{array}$$

When we consider the *n*th extension, a channel is given by:(200)$$\begin{array}{c} P_{B^{n}|A^{n}}\left(b^{n}|a^{n}\right)=P_{{\tilde{X}}^{n}}\left(\pi_{a^{n}}\left(b^{n}\right)\right), \end{array}$$
where the *n*th extension of the group action is defined by $\pi_{a^{n}}\left(b^{n}\right)=\left(\pi_{a_{1}}\left(b_{1}\right),\ldots ,\pi_{a_{n}}\left(b_{n}\right)\right)$.

Similarly, for *n*-fold extension, we can also construct the virtual conditional additive channel. More precisely, for ${\tilde{X}}^{n}∼P_{{\tilde{X}}^{n}}$, we set $Y^{n}=\varpi \left({\tilde{X}}^{n}\right)=\left(\varpi \left({\tilde{X}}_{1}\right),\ldots ,\varpi \left({\tilde{X}}_{n}\right)\right)$ and:(201)$$\begin{array}{c} P_{X^{n},Y^{n}}\left(x^{n},y^{n}\right):=\frac{P_{{\tilde{X}}^{n}}\left(\pi_{a^{n}}\left(0_{y^{n}}\right)\right)}{\left|\{{a^{\prime }}^{n}\in A^{n}|\pi_{a^{n}}\left(0_{y^{n}}\right)=\pi_{{a^{\prime }}^{n}}\left(0_{y^{n}}\right)\}\right|}. \end{array}$$

### 4.3. Conversion of the BPSK-AWGN Channel into the Conditional Additive Channel

Although we only considered finite input/output sources and channels throughout the paper, in order to demonstrate the utility of the conditional additive channel framework, let us consider the additive white Gaussian noise (AWGN) channel with binary phase shift keying (BPSK) in this section. Let A={0,1} be the input alphabet of the channel, and let B=R be the output alphabet of the channel. For an input a∈A and Gaussian noise *Z* with mean zero and variance $\sigma^{2}$, the output of the channel is given by $B={\left(-1\right)}^{a}+Z$. Then, the conditional probability density function of this channel is given as:(202)$$\begin{array}{c} P_{B|A}\left(b|a\right)=\frac{1}{\sqrt{2\pi }\sigma }e^{-\frac{{\left(b-{\left(-1\right)}^{a}\right)}^{2}}{\sigma^{2}}}. \end{array}$$

Now, to define a conditional additive channel, we choose $Y:=R_{+}$ and define the probability density function $p_{Y}$ on Y with respect to the Lebesgue measure and the conditional distribution $P_{X|Y}\left(x|y\right)$ as:(203)$$\begin{array}{cc} p_{Y}\left(y\right) & :=\frac{1}{\sqrt{2\pi }\sigma }\left(e^{-\frac{{\left(y-1\right)}^{2}}{\sigma^{2}}}+e^{-\frac{{\left(y+1\right)}^{2}}{\sigma^{2}}}\right) \end{array}$$(204)$$\begin{array}{cc} P_{X|Y}\left(0|y\right) & :=\frac{e^{-\frac{{\left(y-1\right)}^{2}}{\sigma^{2}}}}{e^{-\frac{{\left(y-1\right)}^{2}}{\sigma^{2}}}+e^{-\frac{{\left(y+1\right)}^{2}}{\sigma^{2}}}} \end{array}$$(205)$$\begin{array}{cc} P_{X|Y}\left(1|y\right) & :=\frac{e^{-\frac{{\left(y+1\right)}^{2}}{\sigma^{2}}}}{e^{-\frac{{\left(y-1\right)}^{2}}{\sigma^{2}}}+e^{-\frac{{\left(y+1\right)}^{2}}{\sigma^{2}}}} \end{array}$$
for $y\in R_{+}$. When we define $b:={\left(-1\right)}^{x}y\in R$ for x∈{0,1} and $y\in R_{+}$, we have:(206)$$\begin{array}{c} p_{XY|A}\left(y,x|a\right)=\frac{1}{\sqrt{2\pi }\sigma }e^{-\frac{{\left(y-{\left(-1\right)}^{a+x}\right)}^{2}}{\sigma^{2}}}=\frac{1}{\sqrt{2\pi }\sigma }e^{-\frac{{\left({\left(-1\right)}^{x}y-{\left(-1\right)}^{a}\right)}^{2}}{\sigma^{2}}}=\frac{1}{\sqrt{2\pi }\sigma }e^{-\frac{{\left(b-{\left(-1\right)}^{a}\right)}^{2}}{\sigma^{2}}}. \end{array}$$

The relations (Equation 202) and (Equation 206) show that the AWGN channel with BPSK is given as a conditional additive channel in the above sense.

By noting this observation, as explained in Remark 5, the single-shot achievability bounds in Section 3.2 are also valid for continuous *Y*. Furthermore, the discussions for the single-shot converse bounds in Section 4.5 hold even for continuous *Y*. Therefore, the bounds in Section 4.4 and Section 4.5 are also applicable to the BPSK-AWGN channel.

In particular, in the *n* memoryless extension of the BPSK-AWGN channel, the information measures for the noise distribution are given as *n* times the single-shot information measures for the noise distribution. Even in this case, the upper and lower bounds in Section 4.4 and Section 4.5 are also applicable by replacing the information measures by *n* times the single-shot information measures. Therefore, we obtain finite-length upper and lower bounds of the optimal coding length for the memoryless BPSK-AWGN channel. Furthermore, even though the additive noise is not Gaussian, when the probability density function $p_{Z}$ of the additive noise *Z* satisfies the symmetry $p_{Z}\left(z\right)=p_{Z}\left(-z\right)$, the BPSK channel with the additive noise *Z* can be converted to a conditional additive channel in the same way.

### 4.4. Achievability Bound Derived by Source Coding with Side-Information

In this subsection, we give a code for a conditional additive channel from a code of source coding with side-information in a canonical way. In this construction, we see that the decoding error probability of the channel code equals that of the source code.

When the channel is given as the conditional additive channel with conditional additive noise distribution $P_{X^{n}Y^{n}}$ as (Equation 186) and X=A is the finite field $F_{q}$, we can construct a linear channel code from a source code with full side-information whose encoder and decoder are $f_{n}$ and $d_{n}$ as follows. First, we assume linearity for the source encoder $f_{n}$. Let $C_{n}\left(f_{n}\right)$ be the kernel of the linear encoder $f_{n}$ of the source code. Suppose that the sender sends a codeword $c_{n}\in C_{n}\left(f_{n}\right)$ and $\left(c_{n}+X^{n},Y^{n}\right)$ is received. Then, the receiver computes the syndrome $f_{n}\left(c_{n}+X^{n}\right)=f_{n}\left(c_{n}\right)+f_{n}\left(X^{n}\right)=f_{n}\left(X^{n}\right)$, estimates $X^{n}$ from $f_{n}\left(X^{n}\right)$ and $Y^{n}$, and subtracts the estimate from $c_{n}+X^{n}$. That is, we choose the channel decoder ${\tilde{d}}_{n}$ as:(207)$$\begin{array}{c} {\tilde{d}}_{n}\left(x^{\prime n},y^{n}\right):=x^{\prime n}-d_{n}\left(f_{n}\left(x^{\prime n}\right),y^{n}\right). \end{array}$$

We succeed in decoding in this channel coding if and only if $d_{n}\left(f_{n}\left(X^{n}\right),Y^{n}\right)$ equals $X^{n}$. Thus, the error probability of this channel code coincides with that of the source code for the correlated source $\left(X^{n},Y^{n}\right)$. In summary, we have the following lemma, which was first pointed out in [27].

**Lemma** **19**([27], (19)). *Given a linear encoder $f_{n}$ and a decoder $d_{n}$ for a source code with side-information with distribution $P_{X^{n}Y^{n}}$, let $I_{\mathrm{Ker}(f_{n})}$ and ${\tilde{d}}_{n}$ be the channel encoder and decoder induced by $\left(f_{n},d_{n}\right)$. Then, the error probability of channel coding for the conditionally additive channel with noise distribution $P_{X^{n}Y^{n}}$ satisfies:*
(208)$$\begin{array}{c} P_{c}^{\left(n\right)}\left[\left(I_{\mathrm{Ker}(f_{n})},{\tilde{d}}_{n}\right)|P_{X^{n}Y^{n}}\right]=P_{s}^{\left(n\right)}\left[\left(f_{n},d_{n}\right)|P_{X^{n}Y^{n}}\right]. \end{array}$$
*Furthermore, (in fact, when we additionally impose the linearity on the random function F in the definition (114) for the definition of ${\bar{P}}_{s}\left(M|P_{X^{n}Y^{n}}\right)$, the result in [27] implies that the equality in (Equation 209) holds) taking the infimum for $F_{n}$ chosen to be a linear two-universal hash function, we also have:*
(209)$$\begin{array}{cc} \; & {\bar{P}}_{c}\left(n,k\right)=\underset{F_{n}\in F_{l}}{\sup}E_{F_{n}}\left[P_{c}^{\left(n\right)}\left[\Psi \left(F_{n}\right)\right]\right]\leq \underset{F_{n}\in F_{l}}{\sup}E_{F_{n}}\left[P_{c}^{\left(n\right)}\left[\left(I_{\mathrm{Ker}(F_{n})},{\tilde{d}}_{n}\right)\right]\right]\\ = & \underset{F_{n}\in F_{l}}{\sup}E_{F_{n}}P_{s}^{\left(n\right)}\left[\left(F_{n},d_{n}\right)\right]\leq \underset{F_{n}\in F}{\sup}E_{F_{n}}P_{s}^{\left(n\right)}\left[\left(F_{n},d_{n}\right)\right]={\bar{P}}_{s}^{\left(n\right)}\left(\right|A^{n-k}\left|\right). \end{array}$$


By using this observation and the results in Section 3.2, we can derive the achievability bounds. By using the conversion argument in Section 4.2, we can also construct a channel code for a regular channel from a source code with full side-information. Although the following bounds are just a specialization of known bounds for conditional additive channels, we review these bounds here to clarify the correspondence between the bounds in source coding with side-information and channel coding.

From Lemma 13 and (Equation 209), we have the following.

**Lemma** **20**([2]). *The following bound holds:*
(210)$$\begin{array}{c} {\bar{P}}_{c}\left(n,k\right)\leq \underset{\gamma \geq 0}{\inf}\left[P_{X^{n}Y^{n}}\left\{\log\frac{1}{P_{X^{n}|Y^{n}}\left(x^{n}|y^{n}\right)}>\gamma \right\}+\frac{e^{\gamma }}{{\left|A\right|}^{n-k}}\right]. \end{array}$$

From Lemma 14 and (Equation 209), we have the following exponential-type bound.

**Lemma** **21**([6]). *The following bound holds:*
(211)$$\begin{array}{c} {\bar{P}}_{c}\left(n,k\right)\leq \underset{-\frac{1}{2}\leq \theta \leq 0}{\inf}{\left|A\right|}^{\frac{\theta (n-k)}{1+\theta }}e^{-\frac{\theta }{1+\theta }H_{1+\theta }^{↑}\left(X^{n}|Y^{n}\right)}. \end{array}$$

From Lemma 15 and (Equation 209), we have the following slightly loose exponential bound.

**Lemma** **22**([3,70]). *The following bound holds (The bound (Equation 212) was derived in the original Japanese edition of [3], but it is not written in the English edition [3]. The quantum analogue was derived in [70].):*
(212)$$\begin{array}{c} {\bar{P}}_{c}\left(n,k\right)\leq \underset{-1\leq \theta \leq 0}{\inf}{\left|A\right|}^{\theta (n-k)}e^{-\theta H_{1+\theta }^{↓}\left(X^{n}|Y^{n}\right)}. \end{array}$$

When *X* has no side-information, i.e., the virtual channel is additive, we have the following special case of Lemma 21.

**Lemma** **23**([6]). *Suppose that X has no side-information. Then, the following bound holds:*
(213)$$\begin{array}{c} {\bar{P}}_{c}\left(n,k\right)\leq \underset{-\frac{1}{2}\leq \theta \leq 0}{\inf}{\left|A\right|}^{\frac{\theta (n-k)}{1+\theta }}e^{-\frac{\theta }{1+\theta }H_{1+\theta }\left(X^{n}\right)}. \end{array}$$

### 4.5. Converse Bound

In this subsection, we show some converse bounds. The following is the information spectrum-type converse shown in [4].

**Lemma** **24**([4], Lemma 4). *For any code $\Psi_{n}=\left(e_{n},d_{n}\right)$ and any output distribution $Q_{B^{n}}\in P\left(B^{n}\right)$, we have:*
(214)$$\begin{array}{c} P_{c}^{\left(n\right)}\left[\Psi_{n}\right]\geq \underset{\gamma \geq 0}{\sup}\left[\sum_{m=1}^{M_{n}}\frac{1}{M_{n}}P_{B^{n}|A^{n}}\left\{\log\frac{P_{B^{n}|A^{n}}\left(b^{n}|e_{n}\left(m\right)\right)}{Q_{B^{n}}\left(b^{n}\right)}<\gamma \right\}-\frac{e^{\gamma }}{M_{n}}\right]. \end{array}$$

When a channel is a conditional additive channel, we have:(215)$$\begin{array}{c} P_{B^{n}|A^{n}}\left(a^{n}+x^{n},y^{n}|a^{n}\right)=P_{X^{n}Y^{n}}\left(x^{n},y^{n}\right). \end{array}$$

By taking the output distribution $Q_{B^{n}}$ as:(216)$$\begin{array}{c} Q_{B^{n}}\left(a^{n}+x^{n},y^{n}\right)=\frac{1}{{\left|A\right|}^{n}}Q_{Y^{n}}\left(y^{n}\right) \end{array}$$
for some $Q_{Y^{n}}\in P\left(Y^{n}\right)$, as a corollary of Lemma 24, we have the following bound.

**Lemma** **25.**
*When a channel is a conditional additive channel, for any distribution $Q_{Y^{n}}\in P\left(Y^{n}\right)$, we have:*
(217)$$\begin{array}{c} P_{c}^{\left(n\right)}\left(M_{n}\right)\geq \underset{\gamma \geq 0}{\sup}\left[P_{X^{n}Y^{n}}\left\{\log\frac{Q_{Y^{n}}\left(y^{n}\right)}{P_{X^{n}Y^{n}}\left(x^{n},y^{n}\right)}>n\log\left|A\right|-\gamma \right\}-\frac{e^{\gamma }}{M_{n}}\right]. \end{array}$$


**Proof.** By noting (Equation 215) and (Equation 216), the first term of the right-hand side of (Equation 214) can be rewritten as:
(218)$$\begin{array}{cc} \; & \sum_{m=1}^{M_{n}}\frac{1}{M_{n}}P_{B^{n}|A^{n}}\left\{\log\frac{P_{B^{n}|A^{n}}\left(b^{n}|e_{n}\left(m\right)\right)}{Q_{B^{n}}\left(b^{n}\right)}<\gamma \right\} \end{array}$$
(219)$$\begin{array}{cc} \; & =\sum_{m=1}^{M_{n}}\frac{1}{M_{n}}P_{X^{n}Y^{n}}\left\{\log\frac{P_{B^{n}|A^{n}}\left(e_{n}\left(m\right)+x^{n},y^{n}|e_{n}\left(m\right)\right)}{Q_{B^{n}}\left(e_{n}\left(m\right)+x^{n},y^{n}\right)}\right\} \end{array}$$
(220)$$\begin{array}{cc} \; & =P_{X^{n}Y^{n}}\left\{\log\frac{Q_{Y^{n}}\left(y^{n}\right)}{P_{X^{n}Y^{n}}\left(x^{n},y^{n}\right)}>n\log\left|A\right|-\gamma \right\}, \end{array}$$
which implies the statement of the lemma. □

A similar argument as in Theorem 5 also derives from the following converse bound.

**Theorem** **16.**
*For any $Q_{Y^{n}}\in P\left(Y^{n}\right)$, we have:*
(221)−logPc(n)(Mn)≤infs>0θ˜∈R,ϑ≥0[(1+s)θ˜H1+θ˜(PXnYn|QYn)−H1+(1+s)θ˜(PXnYn|QYn)
(222)−(1+s)log1−2e−−ϑR+(θ˜+ϑ(1+θ˜))H1+θ˜+ϑ(1+θ˜)(PXnYn|QYn)−(1+ϑ)θ˜H1+θ˜(PXnYn|QYn)1+ϑ]/s≤infs>0−1<θ˜<θ(a(R))[(1+s)θ˜H1+θ˜(PXnYn|QYn)−H1+(1+s)θ˜(PXnYn|QYn)
(223)$$\begin{array}{c} -\left(1+s\right)\log\left(1-2e^{\left(\theta \left(a\left(R\right)\right)-\tilde{\theta }\right)a\left(R\right)-\theta \left(a\left(R\right)\right)H_{1+\theta (a(R))}\left(P_{X^{n}Y^{n}}|Q_{Y^{n}}\right)+\tilde{\theta }H_{1+\tilde{\theta }}\left(P_{X^{n}Y^{n}}|Q_{Y^{n}}\right)}\right)]/s, \end{array}$$
*where $R=n\log|A|-\log M_{n}$, and θ(a) and a(R) are the inverse functions defined in (Equation 29) and (Equation 32), respectively.*


**Proof.** See Appendix L. □

### 4.6. Finite-Length Bound for the Markov Noise Channel

From this section, we address the conditional additive channel whose conditional additive noise is subject to the Markov chain. Here, the input alphabet $A^{n}$ equals the additive group $X^{n}=F_{q}^{n}$, and the output alphabet $B^{n}$ is $X^{\times }Y^{n}$. That is, the transition matrix describing the channel is given by using a transition matrix *W* on $X^{\times }Y^{n}$ and an initial distribution *Q* as:(224)$$\begin{array}{c} P_{B^{n}|A^{n}}\left(x^{n}+a^{n},y^{n}|a^{n}\right)=Q\left(x_{1},y_{1}\right)\prod_{i=2}^{n}W\left(x_{i},y_{i}|x_{i-1},y_{i-1}\right). \end{array}$$

As in Section 2.2, we consider two assumptions on the transition matrix *W* of the noise process (X,Y), i.e., Assumptions 1 and 2. We also use the same notations as in Section 2.2.

**Example** **6**(Gilbert–Elliot channel with state-information available at the receiver). *The Gilbert–Elliot channel [29,30] is characterized by a channel state $Y^{n}$ on $Y^{n}={\left\{0,1\right\}}^{n}$ and an additive noise $X^{n}$ on $X^{n}={\left\{0,1\right\}}^{n}$. The noise process $\left(X^{n},Y^{n}\right)$ is a Markov chain induced by the transition matrix W introduced in Example 3. For the channel input $a^{n}$, the channel output is given by $\left(a^{n}+X^{n},Y^{n}\right)$ when the state-information is available at the receiver. Thus, this channel can be regarded as a conditional additive channel, and the transition matrix of the noise process satisfies Assumption 2.*

Proofs of the following bounds are almost the same as those in Section 3.3, and thus omitted. The combination of Lemmas 10 and 22 derives the following achievability bound.

**Theorem** **17**(Direct, Ass. 1). *Suppose that the transition matrix W of the conditional additive noise satisfies Assumption 1. Let $R:=\frac{n-k}{n}\log\left|A\right|$. Then, we have:*
(225)$$\begin{array}{ccc} -\log{\bar{P}}_{c}\left(n,k\right) & \geq & \underset{-1\leq \theta \leq 0}{\sup}\left[-\theta nR+\left(n-1\right)\theta H_{1+\theta }^{↓,W}\left(X|Y\right)+\underline{\delta }\left(\theta \right)\right]. \end{array}$$

Theorem 16 for $Q_{Y^{n}}=P_{Y^{n}}$ and Lemma 10 yield the following converse bound.

**Theorem** **18**(Converse, Ass. 1). *Suppose that transition matrix W of the conditional additive noise satisfies Assumption 1. Let $R:=\log|A|-\frac{1}{n}\log M_{n}$. If $H^{W}\left(X|Y\right)<R<H_{0}^{↓,W}\left(X|Y\right)$, then we have:*
(226)−logPc(n)(Mn)≤infs>0−1<θ˜<θ(a(R))[(n−1)(1+s)θ˜H1+θ˜↓,W(X|Y)−H1+(1+s)θ˜↓,W(X|Y)+δ1
(227)$$\begin{array}{c} -\left(1+s\right)\log\left(1-2e^{\left(n-1\right)\left[\left(\theta \left(a\left(R\right)\right)-\tilde{\theta }\right)a\left(R\right)-\theta \left(a\left(R\right)\right)H_{1+\theta (a(R))}^{↓,W}\left(X|Y\right)+\tilde{\theta }H_{1+\tilde{\theta }}^{↓,W}\left(X|Y\right)\right]+\delta_{2}}\right)]/s, \end{array}$$
*where $\theta \left(a\right)=\theta^{↓}\left(a\right)$ and $a\left(R\right)=a^{↓}\left(R\right)$ are the inverse functions defined by (Equation 67) and (Equation 70), respectively, and:*
(228)$$\begin{array}{ccc} \delta_{1} & := & \left(1+s\right)\bar{\delta }\left(\tilde{\theta }\right)-\underline{\delta }\left(\left(1+s\right)\tilde{\theta }\right), \end{array}$$
(229)$$\begin{array}{ccc} \delta_{2} & := & \frac{\left(\theta \left(a\left(R\right)\right)-\tilde{\theta }\right)R-\left(1+\tilde{\theta }\right)\underline{\delta }\left(\theta \left(a\left(R\right)\right)\right)+\left(1+\theta \left(a\left(R\right)\right)\right)\bar{\delta }\left(\tilde{\theta }\right)}{1+\theta (a(R))}. \end{array}$$

Next, we derive tighter bounds under Assumption 2. From Lemmas 11 and 21, we have the following achievability bound.

**Theorem** **19**(Direct, Ass. 2). *Suppose that the transition matrix W of the conditional additive noise satisfies Assumption 2. Let $R:=\frac{n-k}{n}\log\left|A\right|$. Then, we have:*
(230)$$\begin{array}{c} -\log{\bar{P}}_{c}\left(n,k\right)\geq \underset{-\frac{1}{2}\leq \theta \leq 0}{\sup}\frac{-\theta nR+\left(n-1\right)\theta H_{1+\theta }^{↑,W}\left(X|Y\right)}{1+\theta }+\underline{\xi }\left(\theta \right). \end{array}$$

By using Theorem 16 for $Q_{Y^{n}}=P_{Y^{n}}^{\left(1+\theta (a(R))\right)}$ and Lemma 12, we obtain the following converse bound.

**Theorem** **20**(Converse, Ass. 2). *Suppose that the transition matrix W of the conditional additive noise satisfies Assumption 2. Let $R:=\log|A|-\frac{1}{n}\log M_{n}$. If $H^{W}\left(X|Y\right)<R<H_{0}^{↑,W}\left(X|Y\right)$, we have:*
(231)−logPc(n)(Mn)≤infs>0−1<θ˜<θ(a(R))[(n−1)(1+s)θ˜H1+θ˜,1+θ(a(R))W(X|Y)−H1+(1+s)θ˜,1+θ(a(R))W(X|Y)+δ1
(232)$$\begin{array}{c} -\left(1+s\right)\log\left(1-2e^{\left(n-1\right)\left[\left(\theta \left(a\left(R\right)\right)-\tilde{\theta }\right)a\left(R\right)-\theta \left(a\left(R\right)\right)H_{1+\theta (a(R))}^{↑,W}\left(X|Y\right)+\tilde{\theta }H_{1+\tilde{\theta },1+\theta \left(a\left(R\right)\right)}^{W}\left(X|Y\right)\right]+\delta_{2}}\right)]/s, \end{array}$$
*where $\theta \left(a\right)=\theta^{↑}\left(a\right)$ and $a\left(R\right)=a^{↑}\left(R\right)$ are the inverse functions defined by (Equation 71) and (Equation 73), respectively, and:*
(233)$$\begin{array}{ccc} \delta_{1} & := & \left(1+s\right)\bar{\zeta }\left(\tilde{\theta },\theta \left(a\left(R\right)\right)\right)-\underline{\zeta }\left(\left(1+s\right)\tilde{\theta },\theta \left(a\left(R\right)\right)\right), \end{array}$$
(234)$$\begin{array}{ccc} \delta_{2} & := & \frac{\left(\theta \left(a\left(R\right)\right)-\tilde{\theta }\right)R-\left(1+\tilde{\theta }\right)\underline{\zeta }\left(\theta \left(a\left(R\right)\right),\theta \left(a\left(R\right)\right)\right)+\left(1+\theta \left(a\left(R\right)\right)\right)\bar{\zeta }\left(\tilde{\theta },\theta \left(a\left(R\right)\right)\right)}{1+\theta (a(R))}. \end{array}$$

Finally, when *X* has no side-information, i.e., the channel is additive, we obtain the following achievability bound from Lemma 23.

**Theorem** **21**(Direct, no-side-information). *Let $R:=\frac{n-k}{n}\log\left|A\right|$. Then, we have:*
(235)$$\begin{array}{c} -\log{\bar{P}}_{c}\left(n,k\right)\geq \underset{-\frac{1}{2}\leq \theta \leq 0}{\sup}\frac{-\theta nR+\left(n-1\right)\theta H_{1+\theta }^{W}\left(X\right)+\underline{\delta }\left(\theta \right)}{1+\theta }. \end{array}$$

**Remark** **9.**
*Our treatment for the Markov conditional additive channel covers Markov regular channels because Markov regular channels can be reduced to Markov conditional additive channels as follows. Let $\tilde{X}={\left\{{\tilde{X}}^{n}\right\}}_{n=1}^{\infty }$ be a Markov chain on B whose distribution is given by:*
(236)$$\begin{array}{c} P_{{\tilde{X}}^{n}}\left({\tilde{x}}^{n}\right)=Q\left({\tilde{x}}_{1}\right)\prod_{i=2}^{n}\tilde{W}\left({\tilde{x}}_{i}|{\tilde{x}}_{i-1}\right) \end{array}$$
*for a transition matrix $\tilde{W}$ and an initial distribution Q. Let $\left(X,Y\right)={\left\{\left(X^{n},Y^{n}\right)\right\}}_{n=1}^{\infty }$ be the noise process of the conditional additive channel derived from the noise process $\tilde{X}$ of the regular channel by the argument of Section 4.2. Since we can write:*
(237)$$\begin{array}{c} P_{X^{n}Y^{n}}\left(x^{n},y^{n}\right)=Q\left(\iota_{y_{1}}^{-1}\left(\vartheta_{y_{1}}\left(x_{1}\right)\right)\right)\frac{1}{\left|\mathrm{Stb}(0_{y_{1}})\right|}\prod_{i=2}^{n}\tilde{W}\left(\iota_{y_{i}}^{-1}\left(\vartheta_{y_{i}}\left(x_{i}\right)\right)|\iota_{y_{i-1}}^{-1}\left(\vartheta_{y_{i-1}}\left(x_{i-1}\right)\right)\right)\frac{1}{\left|\mathrm{Stb}(0_{y_{i}})\right|}, \end{array}$$
*the process (X,Y) is also a Markov chain. Thus, the regular channel given by $\tilde{X}$ is reduced to the conditional additive channel given by (X,Y).*


### 4.7. Second-Order

To discuss the asymptotic performance, we introduce the quantity:(238)$$\begin{array}{c} C:=\log|A|-H^{W}\left(X|Y\right). \end{array}$$

By applying the central limit theorem (cf. [67] (Theorem 27.4, Example 27.6)) to Lemmas 20 and 25 for $Q_{Y^{n}}=P_{Y^{n}}$, and by using Theorem 2, we have the following.

**Theorem** **22.**
*Suppose that the transition matrix W of the conditional additive noise satisfies Assumption 1. For arbitrary ε∈(0,1), we have:*
(239)$$\begin{array}{c} \log M\left(n,\varepsilon \right)=k\left(n,\varepsilon \right)\log|A|=Cn+\sqrt{V^{W}\left(X|Y\right)}\Phi^{-1}\left(\varepsilon \right)\sqrt{n}+o\left(\sqrt{n}\right). \end{array}$$


**Proof.** This theorem follows in the same manner as the proof of Theorem 11 by replacing Lemma 13 with Lemma 20 (achievability) and Lemma 18 with Lemma 25 (converse). □

From the above theorem, the (first-order) capacity of the conditional additive channel under Assumption 1 is given by:(240)$$\begin{array}{c} \lim_{n\to \infty }\frac{1}{n}\log M\left(n,\varepsilon \right)=\lim_{n\to \infty }\frac{1}{n}\log\frac{k(n,\varepsilon )\log|A|}{n}=C \end{array}$$
for every 0<ε<1. In the next subsections, we consider the asymptotic behavior of the error probability when the rate is smaller than the capacity in the moderate deviation regime and the large deviation regime, respectively.

### 4.8. Moderate Deviation

From Theorems 17 and 18, we have the following.

**Theorem** **23.**
*Suppose that the transition matrix W of the conditional additive noise satisfies Assumption 1. For arbitrary t∈(0,1/2) and δ>0, we have:*
(241)$$\begin{array}{ccc} \lim_{n\to \infty }-\frac{1}{n^{1-2t}}\log P_{c}^{\left(n\right)}\left(e^{nC-n^{1-t}\delta }\right) & = & \lim_{n\to \infty }-\frac{1}{n^{1-2t}}\log{\bar{P}}_{c}^{\left(n\right)}\left(n,\frac{nC-n^{1-t}\delta }{\log|A|}\right) \end{array}$$
(242)$$\begin{array}{cc} = & \frac{\delta^{2}}{2V^{W}\left(X|Y\right)}. \end{array}$$


**Proof.** The theorem follows in the same manner as Theorem 12 by replacing Theorem 6 with Theorem 17 (achievability) and Theorem 8 with Theorem 18 (converse). □

### 4.9. Large Deviation

From Theorem 17 and Theorem 18, we have the following.

**Theorem** **24.**
*Suppose that the transition matrix W of the conditional additive noise satisfies Assumption 1. For $H^{W}\left(X|Y\right)<R$, we have:*
(243)$$\begin{array}{c} \underset{n\to \infty }{lim inf}-\frac{1}{n}\log{\bar{P}}_{c}^{\left(n\right)}\left(n,n\left(1-\frac{R}{\log|A|}\right)\right)\geq \underset{-1\leq \theta \leq 0}{\sup}\left[-\theta R+\theta H_{1+\theta }^{↓,W}\left(X|Y\right)\right]. \end{array}$$

*On the other hand, for $H^{W}\left(X|Y\right)<R<H_{0}^{↓,W}\left(X|Y\right)$, we have:*
(244)$$\begin{array}{ccc} \underset{n\to \infty }{lim sup}-\frac{1}{n}\log P_{c}^{\left(n\right)}\left(e^{n(\log|A|-R)}\right) & \leq & -\theta \left(a\left(R\right)\right)a\left(R\right)+\theta \left(a\left(R\right)\right)H_{1+\theta (a(R))}^{↓,W}\left(X|Y\right) \end{array}$$
(245)$$\begin{array}{cc} = & \underset{-1<\theta \leq 0}{\sup}\frac{-\theta R+\theta H_{1+\theta }^{↓,W}\left(X|Y\right)}{1+\theta }. \end{array}$$


**Proof.** The theorem follows in the same manner as Theorem 13 by replacing Theorem 6 with Theorem 17 (achievability) and Theorem 8 with Theorem 18 (converse). □

Under Assumption 2, from Theorems 19 and 20, we have the following tighter bound.

**Theorem** **25.**
*Suppose that the transition matrix W of the conditional additive noise satisfies Assumption 2. For $H^{W}\left(X|Y\right)<R$, we have:*
(246)$$\begin{array}{c} \underset{n\to \infty }{lim inf}-\frac{1}{n}\log{\bar{P}}_{c}^{\left(n\right)}\left(n,n\left(1-\frac{R}{\log|A|}\right)\right)\geq \underset{-\frac{1}{2}\leq \theta \leq 0}{\sup}\frac{-\theta R+\theta H_{1+\theta }^{↑,W}\left(X|Y\right)}{1+\theta }. \end{array}$$

*On the other hand, for $H^{W}\left(X|Y\right)<R<H_{0}^{↑,W}\left(X|Y\right)$, we have:*
(247)$$\begin{array}{ccc} \underset{n\to \infty }{lim sup}-\frac{1}{n}\log P_{c}^{\left(n\right)}\left(e^{n(\log|A|-R)}\right) & \leq & -\theta \left(a\left(R\right)\right)a\left(R\right)+\theta \left(a\left(R\right)\right)H_{1+\theta (a(R))}^{↑,W}\left(X|Y\right) \end{array}$$
(248)$$\begin{array}{cc} = & \underset{-1<\theta \leq 0}{\sup}\frac{-\theta R+\theta H_{1+\theta }^{↑,W}\left(X|Y\right)}{1+\theta }. \end{array}$$


**Proof.** The theorem follows the same manner as Theorem 14 by replacing Theorem 9 with Theorem 19 and Theorem 10 with Theorem 20. □

When *X* has no side-information, i.e., the channel is additive, from Theorem 21 and (Equation 245), we have the following.

**Theorem** **26.**
*For $H^{W}\left(X\right)<R$, we have:*
(249)$$\begin{array}{c} \underset{n\to \infty }{lim inf}-\frac{1}{n}\log{\bar{P}}_{c}^{\left(n\right)}\left(n,n\left(1-\frac{R}{\log|A|}\right)\right)\geq \underset{-\frac{1}{2}\leq \theta \leq 0}{\sup}\frac{-\theta R+\theta H_{1+\theta }^{W}\left(X\right)}{1+\theta }. \end{array}$$

*On the other hand, for $H^{W}\left(X\right)<R<H_{0}^{W}\left(X\right)$, we have:*
(250)$$\begin{array}{c} \underset{n\to \infty }{lim sup}-\frac{1}{n}\log P_{c}^{\left(n\right)}\left(e^{n(\log|A|-R)}\right)\leq \underset{-1<\theta \leq 0}{\sup}\frac{-\theta R+\theta H_{1+\theta }^{W}\left(X\right)}{1+\theta }. \end{array}$$


**Proof.** The first claim follows by taking the limit of Theorem 21, and the second claim follows as a special case of (Equation 245) without side-information. □

### 4.10. Summary of the Results

The results shown in this section for the Markov conditional additive noise are summarized in Table 3. The check marks 🗸 indicate that the tight asymptotic bounds (large deviation, moderate deviation, and second-order) can be obtained from those bounds. The marks 🗸${}^{*}$ indicate that the large deviation bound can be derived up to the critical rate. The computational complexity “Tail” indicates that the computational complexities of those bounds depend on the computational complexities of tail probabilities. It should be noted that Theorem 18 is derived from a special case ($Q_{Y}=P_{Y}$) of Theorem 16. The asymptotically optimal choice is $Q_{Y}=P_{Y}^{\left(1+\theta \right)}$. Under Assumption 1, we can derive the bound of the Markov case only for that special choice of $Q_{Y}$, while under Assumption 2, we can derive the bound of the Markov case for the optimal choice of $Q_{Y}$. Furthermore, Theorem 18 is not asymptotically tight in the large deviation regime in general, but it is tight if *X* has no side-information, i.e., the channel is additive. It should be also noted that Theorem 20 does not imply Theorem 18 even for the additive channel case since Assumption 2 restricts the structure of transition matrices even when *X* has no side-information.

## 5. Discussion and Conclusions

In this paper, we developed a unified approach to source coding with side-information and channel coding for a conditional additive channel for finite-length and asymptotic analyses of Markov chains. In our approach, the conditional Rényi entropies defined for transition matrices played important roles. Although we only illustrated the source coding with side-information and the channel coding for a conditional additive channel as applications of our approach, it could be applied to some other problems in information theory such as random number generation problems, as shown in another paper [60].

Our obtained results for the source coding with side-information and the channel coding of the conditional additive channel has been extended to the case when the side-information is continuous like the real line and the joint distribution *X* and *Y* is memoryless. Since this case covers the BPSK-AWGN channel, it can be expected that it covers the MPSK-AWGN channel. Since such channels are often employed in the real channel coding, it is an interesting future topic to investigate the finite-length bound for these channels. Further, we could not define the conditional Rényi entropy for transition matrices of continuous *Y*. Hence, our result could not be extended to such a continuous case. It is another interesting future topic to extend the obtained result to the case with continuous *Y*.

## Figures and Tables

**Figure 1 entropy-22-00460-f001:**
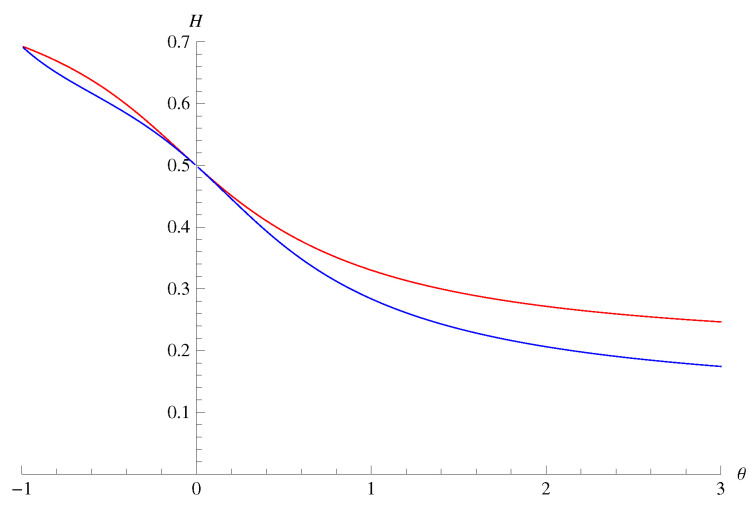
A comparison of $H_{1+\theta }^{↑,W}\left(X|Y\right)$ (upper red curve) and $H_{1+\theta }^{↓,W}\left(X|Y\right)$ (lower blue curve) for the transition matrix of Example 3 with $q_{0}=q_{1}=0.1$, $p_{0}=0.1$, and $p_{1}=0.4$. The horizontal axis is θ, and the vertical axis is the values of the information measures (nats).

**Figure 2 entropy-22-00460-f002:**
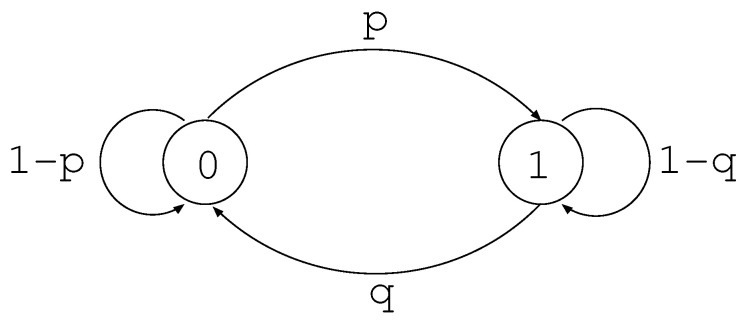
The description of the transition matrix in (Equation 173).

**Figure 3 entropy-22-00460-f003:**
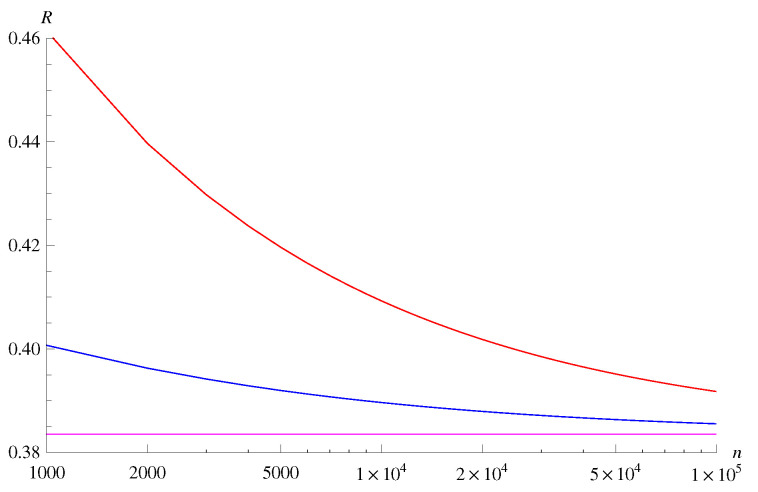
A comparison of the bounds for p=0.1, q=0.2, and $\varepsilon =10^{-3}$. The horizontal axis is the block length *n*, and the vertical axis is the rate *R* (nats). The upper red curve is the achievability bound in Theorem 7. The middle blue curve is the converse bound in Theorem 8. The lower purple line is the first-order asymptotics given by the entropy $H^{W}\left(X\right)$.

**Figure 4 entropy-22-00460-f004:**
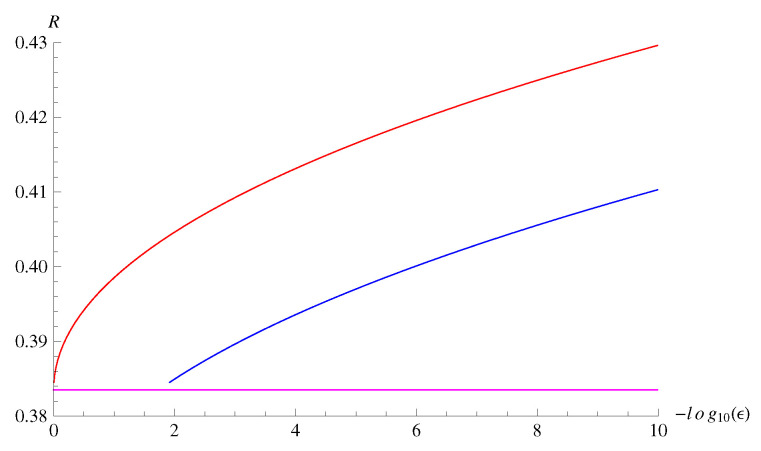
A comparison of the bounds for p=0.1, q=0.2, and n= 10,000. The horizontal axis is $-\log_{10}\left(\varepsilon \right)$, and the vertical axis is the rate *R* (nats). The upper red curve is the achievability bound in Theorem 7. The middle blue curve is the converse bound in Theorem 8. The lower purple line is the first-order asymptotics given by the entropy $H^{W}\left(X\right)$.

**Figure 5 entropy-22-00460-f005:**
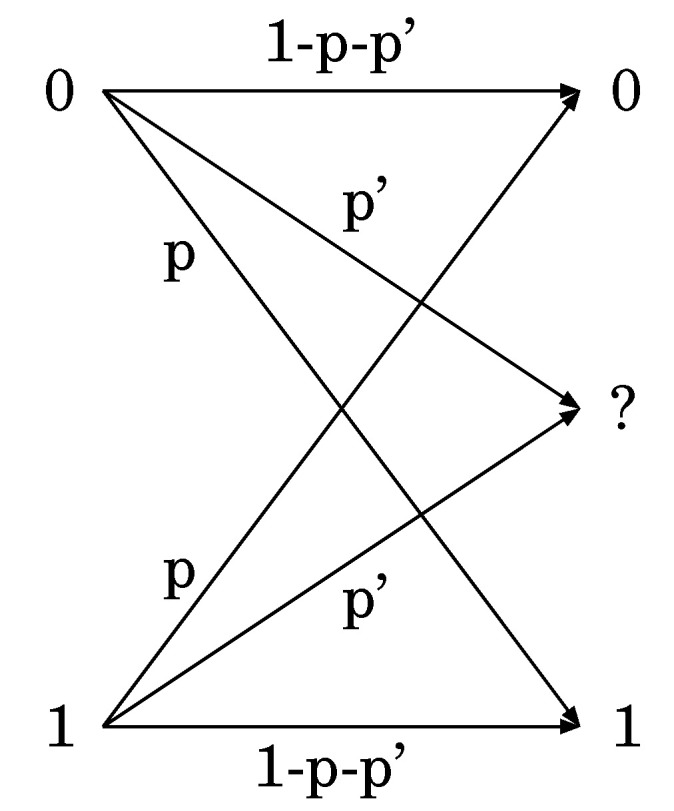
The binary erasure symmetric channel.

**Table 1 entropy-22-00460-t001:** Summary of asymptotic results and finite-length bounds to derive asymptotic results under Assumptions 1 and 2, which are abbreviated to Ass. 1 and Ass. 2.

Problem	First-Order	Large Deviation	Moderate Deviation	Second-Order
SC with SI	Solved (Ass. 1)	${\mathrm{Solved}}^{*}$ (Ass. 2)	Solved (Ass. 1),	Solved (Ass. 1)
		O(1)	O(1)	Tail
CC for Conditional	Solved (Ass. 1)	${\mathrm{Solved}}^{*}$ (Ass. 2)	Solved (Ass. 1)	Solved (Ass. 1)
Additive Channels		O(1)	O(1)	Tail

**Table 2 entropy-22-00460-t002:** Summary of the bounds for source coding with full side-information. No-side means the case with no side-information.

Ach./Conv.	Markov	Single-Shot	$P_{s}$/${\bar{P}}_{s}$	Complexity	Large Deviation	Moderate Deviation	Second Order
Achievability	Theorem 6 (Ass. 1)	Lemma 15	${\bar{P}}_{s}$	O(1)		🗸	
	Theorem 9 (Ass. 2)	Lemma 14	${\bar{P}}_{s}$	O(1)	🗸 ${}^{*}$	🗸	
	Theorem 7 (No-side)	Lemma 16	${\bar{P}}_{s}$	O(1)	🗸 ${}^{*}$	🗸	
	Lemma 13		${\bar{P}}_{s}$	Tail		🗸	🗸
Converse	Theorem 8 (Ass. 1)	(Theorem 5)	$P_{s}$	O(1)		🗸	
	Theorem 10 (Ass. 2)	Corollary 1	$P_{s}$	O(1)	🗸 ${}^{*}$	🗸	
	Theorem 8 (No-side)	(Theorem 5)	$P_{s}$	O(1)	🗸 ${}^{*}$	🗸	
	Lemma 18		$P_{s}$	Tail		🗸	🗸

**Table 3 entropy-22-00460-t003:** Summary of the finite-length bounds for channel coding.

Ach./Conv.	Markov	Single-Shot	$P_{c}$/${\bar{P}}_{c}$	Complexity	Large Deviation	Moderate Deviation	Second Order
Achievability	Theorem 17 (Ass. 1)	Lemma 22	${\bar{P}}_{c}$	O(1)		🗸	
	Theorem 19 (Ass. 2)	Lemma 21	${\bar{P}}_{c}$	O(1)	🗸 ${}^{*}$	🗸	
	Theorem 21 (Additive)	Lemma 23	${\bar{P}}_{c}$	O(1)	🗸 ${}^{*}$	🗸	
	Lemma 20		${\bar{P}}_{c}$	Tail		🗸	🗸
Converse	Theorem 18 (Ass. 1)	(Theorem 16)	$P_{c}$	O(1)		🗸	
	Theorem 20 (Ass. 2)	Theorem 16	$P_{c}$	O(1)	🗸 ${}^{*}$	🗸	
	Theorem 18 (Additive)	(Theorem 16)	$P_{c}$	O(1)	🗸 ${}^{*}$	🗸	
	Lemma 25		$P_{c}$	Tail		🗸	🗸

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
