# Peer review of "Finite-Length Analyses for Source and Channel Coding on Markov Chains†"

_entropy, 2020, doi:10.3390/e22040460_

Round 1
Reviewer 1 Report
This paper presents a study of finite-length bounds for source coding with side information for Markov sources, and for channel coding for channels with conditional Markovian additive noise. Additionally to a rigorous and complete theoretical derivation of these bounds, this paper also presents a computational study versus asymptotic tightness of the bounds.
GENERAL COMMENTS:
- It is pointed out in the introduction that evaluating finite-length bounds for Markov chains is not easy, and for this reason, two main assumptions are needed. There is a nice explanation about why these two assumptions are fundamental from a theoretical point of view. However, I missed a short explanation about the impact of these two assumptions from a practical point of view, together with a brief summary about case studies, or examples in general, that are still valid under these assumptions.
- Sections 3 and 4 are dedicated for the main results of the paper: the analysis of finite-length bounds for source coding with full side-information, and channel coding. Both sections start with a table that summarizes the results of such sections. In my opinion, the summary would be more informative at the end of the section. At the beginning of the section, the reader still do not know what the theorems and lemmas mean, so it is impossible to know why some of them allow to derive large deviation bounds and why others the second order bounds, for example.
- In the numerical example of Section 3.7, there are a couple of plots showing the accuracies of the achievability bound given in Theorem 7, and the converse bound given in Theorem 8. Also, the authors plot the first order asymptotics. I was wondering how interesting would be, if possible, to include also the evaluation of the approximations obtained from the large deviation asymptotic analysis, the moderate deviation analysis, and the second order analysis.
- I missed a numerical example for the channel coding problem. Would it be feasible to come up with an example for which the bounds can be evaluated for this section, too?
MINOR COMMENTS:
- After presenting (1) and (2), the conditional Rényi entropies are introduced. It is indicated that the upper conditional Rényi entropy is given in (12). I would suggest to do the same with the lower conditional Rényi entropy.
- In lines 89 and 94, the lower and upper conditional Rényi entropies for transition matrices are introduced. Could any intuition be given at this point about why these quantities can be evaluated so they give finite-length bounds under Assumptions 1 and 2?
- Theorem 7, Lemma 16, and Lemma 17 are not included in Table 2. However, Theorem 7 seems to be important, since it is used in the numerical example. Why do these bounds not appear in the table? In Table 3 the only missing one is Lemma 25.
- In Equation (204), I guess that the second line should end with, if x=1?
- Between lines 699 and 701, the sections are not mentioned in order. Is there any reason for this?
SOME TYPOS:
- Line 582: holds
- Line 583: bound
- Eq. (136) overlaps with the equation number
- Line 746: linear
- Line 772: ... the above conditional additive channel "is" given as...
- Below line 828: linear
- Line 1261: Wolf
Reviewer 2 Report
Please see attached pdf for detailed comments.
